# Global Well-posedness and Convergence Analysis of Score-based Generative Models via Sharp Lipschitz Estimates

**Connor Mooney**
Department of Mathematics,
University of California at Irvine,
Irvine, CA 92697, USA
mooneycr@uci.edu

**Zhongjian Wang**[*]
Division of Mathematical Sciences,
Nanyang Technological University
21 Nanyang Link, 637371, Singapore
zhongjian.wang@ntu.edu.sg

**Jack Xin**
Department of Mathematics,
University of California at Irvine,
Irvine, CA 92697, USA
jack.xin@uci.edu

**Yifeng Yu**
Department of Mathematics,
University of California at Irvine,
Irvine, CA 92697, USA
yifengy@uci.edu

## Abstract

We establish global well-posedness and convergence of the score-based generative models (SGM) under general assumptions of initial data for score estimation. *For the smooth case*, we start from a Lipschitz bound of the score function with optimal time length. The optimality is validated by an example whose Lipschitz constant of scores is bounded at initial but blows up in finite time. This necessitates the separation of time scales in conventional bounds for non-log-concave distributions. In contrast, our follow up analysis only relies on a local Lipschitz condition and is valid globally in time. This leads to the convergence of numerical scheme without time separation. *For the non-smooth case,* we show that the optimal Lipschitz bound is $O(1/t)$ in the point-wise sense for distributions supported on a compact, smooth and low-dimensional manifold with boundary.

## 1 Introduction

Diffusion models (DM) have become the state-of-the-art tools lately in generative AI Song & Ermon (2019); Song et al. (2021); Dhariwal & Nichol (2021) such as image synthesis Ho et al. (2022); Gao et al. (2023). DMs first evolve data samples with stochastic differential equation (SDE) to gradually inject Gaussian noise until a Gaussian distribution is reached. Then it approximates the drift in the associated backward (time-reversed) SDE and generate a data sample from Gaussian noise. The drift of the backward SDE contains the gradient of the forward logarithmic density (score) that is estimated by solving a matching problem with deep neural network training. The reversibility concept of SDEs dated back to Kolmogorov's work Kolmogorov (1937) in 1937, and the general score formula was derived by Anderson Anderson (1982) in 1982.

Theoretical study on the convergence of DM generated distribution to the target (data) distribution typically assumes that the data distribution admits a density with respect to Lebesgue measure Lee et al. (2022) among others. By also imposing that the *score of the data distribution is Lipschitz continuous*, the score function of the forward process (the drift in the backward process) is well-behaved (not exploding) as the backward time tends to zero when the desired target sample is to be generated. However, this is not always observed in practice and experimentally the score can blow up Kim et al. (2022). In particular, the explosion occurs at generation if the data distribution satisfies the *manifold hypothesis (MH)* Tenenbaum et al. (2000); Goodfellow et al. (2016) which is verified for image data in Brown et al. (2023). Under MH, Pidstrigach (2022) showed that the limit of the continuous backward process with approximate score is well-defined and that the sample

---

[*]Corresponding author.

distribution shares the same support as the target distribution under the integrability conditions on the error of score matching. Also under MH, Bortoli (2022) found quantitative bounds on the 1-Wasserstein distance between a compact target (data) distribution and the generative distribution of DM by allowing the score function to explode as backward time approaches zero.

Both of the references (Pidstrigach (2022) and Bortoli (2022)), among others (Lee et al. (2022) for Langevin MC, Huang et al. (2024) for ODE flows, Chang et al. (2024) for Föllmer flows), require a (locally) Lipschitz estimate for the score function to ensure the well-posedness of the backward SDEs and the approximation bound of the score matching and sampling process.

**The goal of this paper** is to provide sharp estimates that 1) confirm/improve the score assumptions of the existing convergence theory, 2) give insight for the duration of the forward process so that the backward process is well-defined, and 3) justify practical implementation of the backward process (e.g. early stopping strategies or truncation Kim et al. (2022)).

**Related work**  We are aware of the convergence bound of discrete schemes for backward processes in Chen et al. (2023). Our convergence bound takes the KL chain inequality (Proposition C.3.Chen et al. (2023)) as the building block. While equipped with sharp (local) Lipschitz bounds in the paper, we achieved polynomial complexity of sampling in the general smooth $p_0$ setting without separated regimes of schedule. We are also aware of Bortoli (2022) which provides convergence bound in Wasserstein distance under a singular $p_0$ setting, supported on a compact manifold. Due to the potential singular behaviour of the score, early stopping schedules are employed Kim et al. (2022). Additional related work and comparison are discussed in Remark 3.3 for the Lipschitz bound and in Remark 4.7 for the convergence and complexity bound. Our paper provides sharp Lipschitz bounds of the singularities and therefore insights for the choices of schedules and loss normalization between discretization points. In addition, the Lipschitz bounds hold generally for models sharing the same forward process as OU, for example, the probability flow ODE (Equation (13) in Song et al. (2021)).

**The main contributions of this paper are:**

- *Realistic or sharp point-wise gradient and Hessian estimates* of the score potential function $\log p$ from commonly hypothesized data distributions. See Theorem 3.5.
- The first sharp example demonstrating the loss of Lipschtiz bound of the score function as time gets large even with nice initial data. See Theorem 3.4.
- Well-posedness and convergence of the backward diffusion process up to time zero (the generation time) in the smooth setting without separated regime of discretization. See Theorem 4.4 and Theorem 4.6.
- Characterization of the score (and its derivatives) in the setting of manifold hypothesis. See Theorem 3.8.

The rest of the paper is organized as follows. In Section 2, we first introduce settings of the diffusion model and discretization schemes of the backward process. Later, we present the transformation that relates the Fokker Planck equation with unbounded coefficients (density equation of forward process) to the non-linear Hamilton Jacobi equation and heat equation, which serves as the foundation of the analysis. **The main theoretical results, Hessian estimate of score potential function $\log p$, are listed in Section 3.** Based on these estimates, we establish well-posedness of the continuous backward process and convergence bound of discretization in Section 4. Conclusions are stated in Section 5. The details of the proofs are in the Appendix.

## 2 PRELIMINARIES

### 2.1 BACKGROUND AND SETTING THE STAGE

A large class of generative diffusion models can be analyzed under the SDE framework Song et al. (2021). It consists of two processes: forward and backward. *The forward process*, which relates to training, is an Ornstein-Uhlenbeck (OU) process in $\mathbb{R}^n$ as follows:

$$dX_t = -\frac{1}{2}X_t dt + dW_t, \quad \text{for } t \in [0, T] \tag{1}$$

where $W_t$ is a standard Brownian motion, $T$ is the final time such that the distribution of $X_T$ approximates a normal distribution in $\mathbb{R}^n$, namely $\mathcal{N}(0, I_n)$. The initial distribution $X_0$ follows a target (data) distribution in $\mathbb{R}^n$ during the generative task, denoted as $p_0$. *The backward process*, which relates to generation of new data, is defined as an 'inversion' of forward process (1). More precisely, with time reversal $t' = T - t$,

$$d\tilde{X}_{t'} = \left( \frac{1}{2} \tilde{X}_{t'} + \nabla \log p(T - t', \tilde{X}_{t'}) \right) dt' + d\tilde{W}_{t'} \quad \text{for } t' \in [0, T], \tag{2}$$

where $W_{t'}$ is a standard Brownian motion (not necessarily being the same as $W_t$) and the initial distribution $\tilde{X}_0$ follows $\mathcal{N}(0, I_n)$. The term $\nabla \log p$ is introduced in Eq. (2) such that the marginal distributions of the forward and backward processes are identical Anderson (1982).

To be specific, let $p := p(t, x)$ denote the probability distribution function of the forward process (1), which solves the Fokker Planck equation with Cauchy data $p_0$, namely

$$\begin{cases} \partial_t p = \frac{1}{2} (\nabla \cdot (xp) + \Delta p) \\ p(x, 0) = p_0(x). \end{cases} \tag{3}$$

We also denote $P_t$ ($Q_{t'}$ correspondingly) as the marginal distribution of $X_t$ in (1) ($\tilde{X}_{t'}$ in (2)). Given initial distribution for (2) $Q_0 \sim P_T$, then Anderson (1982): $\forall t, Q_t = P_{T-t}$. Especially, $Q_T = P_0$ so data $\sim P_0$ can be generated by solving (2).

In practice, since no closed form expression of $p_0$ is known, the $p$ in (3) is not analytically available. Thus $\nabla \log p$ is approximated by a neural network $s := s_\theta(t, x)$, where $\theta$ denotes latent variables of neural network and is omitted for simplicity of notation. The approximation is obtained by training the neural network with an $L_2$ score estimation loss, $\forall t \in [0, T]$,

$$\mathbb{E}_{x \sim P_t} ||s_\theta(t, x) - \nabla \log p(t, x)||^2.$$

In the analysis, we assume an $\epsilon_0^2$ bounds for this estimation, see Assumption 2.1.

Given the approximation of score $s_\theta$, we employ the exponential scheme Zhang & Chen (2023) with initial distribution $\mathcal{N}(0, I_n)$. More precisely, let $\delta = t_0 \leq t_1 \leq \cdots \leq t_N = T$ be the discretization points. $\delta = 0$ for the normal setting and $\delta > 0$ for the early-stopping setting. Then with $t'_k = T - t_{N-k}$, the process in the discrete scheme is as follows:

$$d\hat{x}_{t'} = (\frac{1}{2} \hat{x}_{t'} + s_\theta(T - t'_k, \hat{x}_{t'_k})) dt + d\hat{w}_{t'} \quad t' \in [t'_k, t'_{k+1}], \quad k = 0, \cdots, N-1, \tag{4}$$

which admits an explicit solution, with $\mu_k \sim \mathcal{N}(0, I_n)$,

$$\hat{x}_{t'_{k+1}} = e^{\frac{1}{2}(t'_{k+1} - t'_k)} \hat{x}_{t'_k} + 2(e^{\frac{1}{2}(t'_{k+1} - t'_k)} - 1) s_\theta(T - t'_k, \hat{x}_{t'_k}) + \sqrt{e^{(t'_{k+1} - t'_k)} - 1} \mu_k.$$

Due to the limited knowledge of $p_0$ as well as the regularity of $\nabla \log p$, we restrict ourselves to uniform discretization points. Detailed selection is stated in the convergence theorems.

We assume the following bound of score approximation at the discretization points,

**Assumption 2.1.** *Let $t_k$ be the discretization point of the scheme* (4),

$$\frac{1}{T} \sum_{k=1}^{N} (t_k - t_{k-1}) E_{x \sim P_{t_k}} \|\nabla \log p(t_k, x) - s_\theta(t_k, x)\|^2 \leq \epsilon_0^2.$$

## 2.2 Foundational Ideas based on Non-linear Hamilton Jacobi Equation

The foundation of our analysis is investigating the behaviour of $\log p$ as the solution of a non-linear Hamilton Jacobi equation (HJE), which is well known to experts. For reader's convenience, we present it here.

We consider the score potential function[1]

$$q(t, x) = -\log p(t, x) - \frac{|x|^2}{2}$$

---

[1] Here we only consider the transform when the distribution of forward process $P_t$ is absolutely continuous with respect to Lebesgue measure. The transform and our analysis are valid for any $t > 0$ in the general case and up to $t = 0$ when $p_0$ is smooth.

whose spatial gradient becomes the drift (score) in the backward (reverse time denoising and generation) process (2) of the diffusion model. The $q$ function satisfies the following PDE:

$$\begin{cases} \partial_t q - \frac{1}{2}\Delta q + \frac{1}{2}(x \cdot \nabla q + |\nabla q|^2) = 0 \\ q(0, x) = g(x), \end{cases} \tag{5}$$

where $g(x) = -\log p_0(x) - |x|^2/2$, which is the non-Gaussian part of the likelihood function.

To simplify Eq.(5), we make a two step change of variables in time. First, let $\tilde{q}(t, x) = q(t, e^{t/2}x)$, then $\tilde{q}$ solves:

$$\partial_t \tilde{q} = \partial_t q + e^{\frac{t}{2}} x \cdot \nabla q(t, e^{\frac{t}{2}}x) = \frac{e^{-t}}{2}(\Delta \tilde{q} - |\nabla \tilde{q}|^2).$$

Then we consider $\bar{q}(t, x) = \tilde{q}(-\log(1-t), x)$, then $\bar{q}$ solves:

$$\begin{cases} \partial_t \bar{q} = \frac{1}{2}(\Delta \bar{q} - |\nabla \bar{q}|^2) & t \in [0, 1) \\ \bar{q}(0, x) = q_0 \end{cases} . \tag{6}$$

**Remark 2.2.** *By a direct calculation*

$$\bar{q}(t, x) = q\left(-\log(1-t), \frac{1}{\sqrt{1-t}}x\right) \text{ or equivalently, } q(t, x) = \bar{q}(1 - e^{-t}, e^{-t/2}x). \tag{7}$$

*Furthermore,*

$$\nabla q(t, x) = e^{-t/2}\nabla \bar{q}(1 - e^{-t}, e^{-t/2}x) \text{ and, } \nabla^2 q(t, x) = e^{-t}\nabla^2 \bar{q}(1 - e^{-t}, e^{-t/2}x).$$

Lastly, we also define $\bar{p}(t, x) = e^{-\bar{q}(t,x)}$, which satisfies

$$\begin{cases} \partial_t \bar{p} = \frac{1}{2}\Delta \bar{p} & \text{on } (0, 1) \times \mathbb{R}^n \\ \bar{p}(0, x) = h(x) = e^{-g(x)}. \end{cases} . \tag{8}$$

The solution of (8) is given by $\bar{p}(t, x) = \frac{1}{(2\pi t)^{\frac{n}{2}}} \int_{\mathbb{R}^n} e^{\frac{-|x-y|^2}{2t}} e^{-g(y)} \, dy$.

To derive reasonable point-wise estimates of gradients and Hessian of the score function $q(t, x)$ that does not involve $1/t$, we will need the following assumption in relevant results. This assumption also ensures the above integration is well-defined for $t \in [0, 1]$, equivalently the well-posedness of Fokker Planck equation (3) for $t \in [0, \infty)$.

**Assumption 2.3.** *The tail distribution is upper bounded by some Gaussian distribution, i.e,*

$$\log p_0(x) - \log p_0(0) \le \alpha_1 - \frac{1}{2}(1 - \alpha_2)|x|^2$$

*for constants $\alpha_2 < 1$ and $\alpha_1 \in \mathbb{R}$. Without loss of generality we assume $\alpha_2 \ge 0$.*

Recalling definition of $g$, it is equivalent to

$$g(x) - g(0) \ge -\frac{\alpha_2}{2}|x|^2 - \alpha_1, \tag{9}$$

Note that Assumption 2.3 implies that the second order moment of the process is bounded, i.e.,

$$E_{p_0}||X||^2 := M_2 < \infty. \tag{10}$$

Technically speaking, the $g(0)$ could be absorbed into $\alpha_1$ in (9). We put it there just to track possible dependence on the dimension $n$. Similarly, we adopt the following technical assumptions in the relevant results to provide more flexibility to track such dependence.

**Assumption 2.4.** *There exists $x_0$, $\alpha_2 \in [0, 1)$, $\alpha_1 \in \mathbb{R}$ such that*

$$g(x) - g(x_0) - \nabla g(x_0) \cdot (x - x_0) \ge -\frac{\alpha_2}{2}|x - x_0|^2 - \alpha_1 \quad \forall x \in \mathbb{R}^n.$$

In particular, if $g$ attains minimum at some point $x_0$, then the Assumption 2.4 holds with $\alpha_2 = \alpha_1 = 0$. Also, if Assumption 2.4 holds, then Assumption 2.3 holds by adjusting the corresponding $\alpha_2$ and $\alpha_1 \in \mathbb{R}$ depending on $g(0)$ and $x_0$, and vice versa. The notation $(\alpha.)$ is abused for simplicity of subsequent derivations without affecting our estimation for dimension dependency.

**General notations** Throughout this paper, for an $n \times n$ matrix $A$, we use the spectral norm

$$||A||_2 = \max_{\{v \in \mathbb{R}^n : |v|=1\}} |Av| = \text{the largest eigenvalue of } \sqrt{AA^\top}. \tag{11}$$

In particular, for a map $F : \mathbb{R}^n \to \mathbb{R}^n$,

$$||\nabla F||_2 \le L \quad \Leftrightarrow \quad |F(x) - F(y)| \le L|x - y|.$$

We also adopt the following notation when comparing two symmetric (Hessian) matrices,

$$A \preceq B \text{ if } B - A \text{ is semi-positive definite.}$$

So for any symmetric matrix $A$, $||A||_2 \le \sigma \Leftrightarrow -\sigma I_n \preceq A \preceq \sigma I_n$. For a map $u : \mathbb{R}^n \to \mathbb{R}^n$, $D^2 u$ denotes the Hessian matrix of the map.

## 3 SHARP HESSIAN BOUND OF SCORE POTENTIAL FUNCTION

The fundamental question, which is directly related to the well-posedness and convergence rate of the diffusion model Bortoli (2022); Lee et al. (2022), is whether for any $T > 0$, there exists a constant $C_T$ that depends only on $T$ and the initial data such that

$$\sup_{[0,T] \times \mathbb{R}^n} ||D^2 q(t,x)||_2 \le C_T \ ?$$

The short time existence of uniform Hessian bound was known in previous literature (see Chen et al. (2023); Mikulincer & Shenfeld (2024) for instance) when $||D^2 \log p_0||$ is bounded . From both a mathematical and application perspective, a natural question is whether it could be extended to all time. In Section 3.1 we provide the first example that shows the short time existence is optimal in sense of lasting time. Precisely speaking, in the proof of Theorem 3.4, we construct an initial distribution $p_0$ such that the Hessian of $\log p$ loses global bound **right** at the limiting time. Inspired by the counter-example, alternatively in Section 3.2 we provide a locally Lipschitz estimate that lasts for $t \in [0, \infty)$. For the non-smooth case, in Section 3.3, we characterize the singular behaviour of $\log p$ and its derivatives.

### 3.1 HESSIAN ESTIMATE OF SCORE POTENTIAL FUNCTION FOR FINITE TIME

The following short-time uniform Hessian, or similar formulations, have been obtained in some previous works. See Remark 3.3 blow. The primary goal of this section is demonstrate that the associated time threshold is sharp (Theorem 3.4).

**Theorem 3.1.** *Let $M_0$ be a nonnegative number. $g \in C^2(\mathbb{R}^n)^2$.*

*(1) If $D^2 g(x) \preceq M_1 I_n$, then*

$$D^2 q(t,x) \preceq e^{-t} M_1 I_n \quad \text{for all } (t,x) \in [0,\infty) \times \mathbb{R}^n.$$

*(2) If $D^2 g(x) \succeq -M_0 I_n$, then for any $T \in \left[0, -\log(1 - \frac{1}{M_0})\right)$, we have*

$$D^2 q(t,x) \succeq -\frac{M_0}{e^t - M_0(e^t - 1)} I_n \quad \text{for all } (t,x) \in (0,T] \times \mathbb{R}^n.$$

*Note that if $M_0 \le 1$, then $T \in [0,\infty)$.*

The proof is in Section C.1.As an immediate corollary, we have that

**Corollary 3.2.** *Given data distribution $p_0 \in C^2(\mathbb{R}^n)$ follows $-L_1 I \preceq \sup_{x \in \mathbb{R}^n} D^2 \log p_0(x) \preceq L_0 I$. Then we have finite time uniform bound of the Hessian: for any $t \in \left[0, -\log(1 - \frac{1}{L_0+1})\right)$,*

$$\sup_{\mathbb{R}^n} ||D^2 \log p(t,x)||_2 \le C_t.$$

---

[2]The assumption is equivalent to $\log p_0 \in C^2(\mathbb{R}^n)$.

*where*

$$C_t = \max\left(\frac{L_0 + 1}{1 - (L_0 + 1)(e^t - 1)} - 1, \ e^{-t}(L_1 - 1) + 1\right). \tag{12}$$

*Furthermore, if* $-\log p_0(x)$ *is a convex function* $(L_0 \leq 0)$, *the estimate bound is global,*

$$0 \preceq -D^2 \log p(t, x) \preceq (e^{-t} L_1 + (1 - e^{-t})) I_n \quad \textit{for all } (t, x) \in [0, \infty) \times \mathbb{R}^n.$$

**Remark 3.3.** *The convex case has been also discussed in Lee et al. (2021) and it leads to single modal distribution. Similar finite bound was also derived in Lemma C.9 in Chen et al. (2023), which follows directly from the representation formula and the generalized Poincaré inequality for log-concave probability measures. We are also aware of bounds similar to Theorem 3.1(2) obtained in Brownian transport map setting Mikulincer & Shenfeld (2024) based on the representation formula and the Brascamp-Lieb inequality that is related to the generalized Poincaré inequality. In this section, we will present a new proof based on PDE (partial differential equation) method via using the convex envelope as a barrier. Our approach is more robust, which does not rely on the representation formula and could be easily adjusted to more general situations. Results of a similar spirit were also obtained in Kim & Milman (2012), where a generalization of Caffarelli's contraction theorem Caffarelli (2000) is proven, also using a parabolic maximum principle, which is different from our method. We will refer the reader as well to Mikulincer & Shenfeld (2023) and also Conforti (2024), where related questions are investigated from a more probabilistic viewpoint. In addition, we would like to point out that beyond the spatially global Hessian bound in the mentioned references (including Mikulincer & Shenfeld (2024)) that will degenerate in finite time, we also provide local Hessian bound that holds for any given finite interval, see Theorem 3.5.*

Given the crucial role of Hessian bound in estimating convergence rate of diffusion model, an important remaining question was whether the lasting time given in Theorem 3.1 is optimal. The result below is our main contribution in this aspect, which shows that the temporal bound $-\log\left(1 - \frac{1}{M_0}\right)$ in the statements of Theorem 3.1 and Corollary 3.2 is sharp.

**Theorem 3.4** (Loss of Uniform Hessian Bound). *There exists a smooth nonnegative $g$ satisfying assumptions in Theorem 3.1 and Corollary 3.2 ($M_0 = M_1 = 2$) such that the corresponding $q(t, x)$ satisfies*

$$\sup_{x \in \mathbb{R}^n} ||D^2 q(\log 2, x)|| = \sup_{x \in \mathbb{R}^n} ||D^2 \bar{q}(1/2, x)|| = \infty.$$

Note that the number $\frac{1}{2}$ can be changed to any given time by re-scaling the function $\bar{q}(\lambda^2 t, \lambda x)$. The detail of construction is in Section C.2

## 3.2 LOCAL ESTIMATE

The following theorem provides point-wise estimates of the score function, which can be quite useful in dealing with more general situations. Technically speaking, $g(x_0)$ and $Dg(x_0)$ can be absorbed into other parameters. Here we choose to display them to track the dependence of relevant parameters on the dimension $n$.

**Theorem 3.5.** *Suppose that $\bar{p} = \bar{p}(t, x)$ is the solution to heat equation (8). Let $|v|_1 = \max\{|v|, 1\}$ for $v \in \mathbb{R}^n$. Fix $x_0 \in \mathbb{R}^n$.*

*(i) Given Assumption 2.3 and $|\nabla g(x)| \leq \beta_1 |x - x_0| + \beta_2$ for $\beta_1, \beta_2 \geq 0$. Then for all $(t, x) \in [0, 1] \times \mathbb{R}^n$,*

$$|\nabla \bar{q}(t, x)| \leq \frac{3\beta_1}{\sqrt{1 - \alpha_2}} \max\{C_n, \ C_{\beta_1, \alpha_2} |x - x_0|_1\} + \beta_2. \tag{13}$$

*Here the two constants $C_n = 2\sqrt{(n + 3)\log\left(\frac{2(1 + 4\beta_1)}{\sqrt{1 - \alpha_2}}\right) + 4n \log n + \alpha_1 + 1 + \frac{\beta_2^2}{\beta_1}}$ and $C_{\beta_1, \alpha_2} = 3\sqrt{\beta_1} + 1 + \frac{6\alpha_2}{\sqrt{1 - \alpha_2}}$.*

*(ii) Assume $||D^2 g(x)||_2 \leq L$ and Assumption 2.4. Then for all $(t, x) \in [0, 1] \times \mathbb{R}^n$,*

$$||D^2 \bar{q}(t, x)|| \leq \frac{10L^2 + L}{1 - \alpha_2} \max\left\{\tilde{C}_n^2, \ (\tilde{C}_{L, \alpha_2})^2 (|x - x_0 - \nabla g(x_0)|_1)^2\right\}, \tag{14}$$

$$|\partial_t \nabla \bar{q}(t,x)| \le \frac{48L^2 + 2L}{\sqrt{t}(1-\alpha_2)^{\frac{3}{2}}} \max \left\{ \tilde{C}_n^3, \ (\tilde{C}_{L,\alpha_2})^3 (|x - x_0 - \nabla g(x_0)|_1)^3 \right\}. \tag{15}$$

*Here the two constants $\tilde{C}_n = 2\sqrt{(n+3)\log\left(\frac{2(1+4L)}{\sqrt{1-\alpha_2}}\right) + 4n \log n + \alpha_1 + 1}$ and $\tilde{C}_{L,\alpha_2} = 3\sqrt{L} + 1 + \frac{6\alpha_2}{\sqrt{1-\alpha_2}}$.*

The proof is in Section C.3. A simpler case with bounded $\nabla g$ is also discussed. See Remark C.2 for another way to bound $|\partial_t \nabla \bar{q}(t,x)|$ by replacing $\frac{L^2}{\sqrt{t}}$ in (15) by $O(nL^3)$.

As an immediate corollary, we have

**Corollary 3.6.** *Assume $||D^2 g(x)||_2 \le L$. Suppose that Assumption 2.4 holds and there exists $C_0 > 0$ such that, $\alpha_1 \le C_0 n$ and $|\nabla g(x_0)| \le C\sqrt{n}$. Then*

$$||D^2 \bar{q}(t,x)|| \le CL^2 \left( n \log n + L|x - x_0 - \nabla g(x_0)|^2 \right)$$
$$|\partial_t \nabla \bar{q}(t,x)| \le CL^2 \left( (n \log n)^{\frac{3}{2}} + L\sqrt{L}|x - x_0 - \nabla g(x_0)|^3 \right).$$

*Here $C$ is a constant independent of $L$ and $n$.*

Note that the further assumptions of scale relates to normalization in $n$ dimension.

**Remark 3.7.** *Owing to 35 in the proof of Theorem 3.5, under the assumption of corollary 3.6, we have for all $m \in \mathbb{N}$*

$$\mathbb{E}_{p(t,x)}(|x(t)|^m) \le O\left(L^{\frac{m}{2}}(n \log n)^{\frac{m}{2}}\right)$$

*This demonstrates that, if we only care about expectations of powers of $D^2 \bar{q}(t,x)$ or $\partial_t \nabla \bar{q}(t,x)$, $||D^2 \bar{q}(t,x)||_2$ behaves like $O(L^3 n \log n)$ and $|\partial_t \nabla \bar{q}(t,x)|$ behaves like $O\left(L^3 \sqrt{L}(n \log n)^{\frac{3}{2}}\right)$. Note that the point $x_0$ itself plays no role in computing the expectation that is translation invariant in the $x$ variable.*

### 3.3 COMPACTLY SUPPORTED DATA DISTRIBUTIONS

In this section, we look at the situation where the data distribution $p_0$ is a positive measure with compact support which is a typical situation in image generation Bortoli (2022). Due to the manifold hypothesis, the support is typically a low dimension set. In this situation, what is important is the asymptotic estimate as $t \to 0$. Assume $\text{supp}(p_0) = D_0 \subset \overline{B_M(0)}$. The following are two known standard estimates(Bortoli (2022)).

$$(1) \quad |\nabla \bar{q}(t,x)| \le \frac{|x| + M}{t}; \tag{16}$$

$$(2) \quad ||D^2 \bar{q}(t,x)||_2 \le \frac{1}{t} + \frac{M^2}{t^2}.$$

The proof is simple, which will be presented in Section C.4 for reader's convenience. Some steps will be used later. The main challenge is whether the above bounds can be improved in order to derive better convergence rate, for instance, Theorem 3 in Bortoli (2022).

**I.** We first demonstrate $O(1/t)$ bound in (1) above is a typical situation that can not be improved.

Fixing $x$, denote by $\bar{y}_t$ the weighted center of mass: $\bar{y}_t = \frac{\int_{D_0} y e^{\frac{-|x-y|^2}{2t}} d\pi_0(y)}{\hat{p}}$, where for the rest of the proof, we denote $p_0(y)dy$ as $d\pi_0(y)$ and $\hat{p} = \int_{D_0} e^{\frac{-|x-y|^2}{2t}} d\pi_0(y)$. For a "regular" $\pi_0$, as $t \to 0$, we expect the measure $\frac{e^{\frac{-|x-y|^2}{2t}} d\pi_0(y)}{\hat{p}}$ will concentrate on $\{y \in D_0| \ |y - x| = d(x, D_0)\}$. Thus,

$$\lim_{t \to 0} d(\bar{y}_t, \Gamma_x) = 0,$$

where $\Gamma_x$ is the convex hull of $\{y \in D_0| \ |y - x| = d(x, D_0)\}$. Then

$$|\nabla \bar{q}(t,x)| = \frac{|x - \bar{y}_t|}{t} \quad \text{and} \quad \liminf_{t \to 0} t|\nabla \bar{q}(t,x)| \ge d(x, \Gamma_x).$$

So if $x \notin \Gamma_x$ (typical situation for low dimension set $D_0$), then $|D\bar{q}(t,x)| = O(1/t)$. Hence the $1/t$ blow up for the gradient bound is usually inevitable, which matches experimental observations Kim et al. (2022). Accordingly, in real applications, the denoising process might only be traced back to a certain $t_0 > 0$, which is corresponding to an initial condition similar to $p(t_0, x)$.

**II.** We now turn our attention to the Hessian bound $O(1/t^2)$ in (2). According to Theorem 3 in Bortoli (2022), if this bound is improved to $O(1/t)$, a better convergence rate in the Wasserstein metric can be achieved. The following theorem establishes that, in typical scenarios, the Hessian bound is $O(1/t)$ rather than $O(1/t^2)$, with the exception of a small set. Consequently, it might be reasonable in practice to assume a Hessian bound of $O(1/t)$ when analyzing convergence rates. Quantifying how frequently this small set could impact the convergence rate remains a challenging problem due to its complex topological structure in the case of nonconvex $D_0$.

For simplicity and clarity, we assume that $D_0$ is a low-dimensional smooth manifold with boundary, though our results extend to manifolds with lower regularity. To illustrate the sharpness of our conclusion, we provide an example in Example 3.9

**Theorem 3.8.** *For $1 \le d \le n$, assume that $D_0 \subset \mathbb{R}^n$ is a d-dimensional compact smooth manifold with boundary and $\pi_0$ is comparable to the uniform distribution on $D_0$. Then for almost everywhere $x \in \mathbb{R}^n$,*
$$||D^2\bar{q}(t,x)||_2 \le \frac{C_x}{t} \quad \text{for } t \in [0,1].$$
*Here $C_x$ is a constant depending only on $x$ and $D_0$. If $D_0$ is convex, then the above holds for all $x \in \mathbb{R}^n$.*

Proof is in Section C.5. We would like to mention that the $O(1/t)$ bound was also derived in Bortoli (2022) for the very special cases, for instance, when $p_0$ follows a uniform distribution product with a normal distribution on a hypercube.

We will present a smooth non-convex $D_0$ that shows the result of Theorem 3.8 is optimal.

**Example 3.9.** *Let $D_0 \subset \mathbb{R}^2$ be the domain obtained by removing a small square $[0,2] \times [-1,1]$ from the big square $[-2,2]^2$ and then mollifying the corners to make it smooth. Here $O = (0,0)$. The $Y$-shaped region*
$$L = \{x \in \mathbb{R}^2 | \text{ there are more than one } y \text{ such that } |x - y| = d(x, D_0)\}.$$
*We also choose $\pi_0 = \frac{\chi_{D_0}}{|D_0|} dx$, i.e., the uniform distribution on $D_0$, where $|D_0|$ is the area of $D_0$. We have that, $||D^2\bar{q}(t,x)||_2 \ge C_x/t^2$ for $x \in L$ and $t \in (0,1]$.*

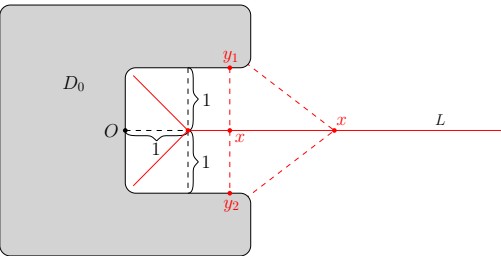

Figure 1: In the above picture, $\Gamma_x = \{sy_1 + (1-s)y_2 : s \in [0,1]\}$

For reader's convenience, we will verify the above when $x = (\theta, 0)$ for $\theta > 1$ in Section C.6. The other parts are left to interested readers as an exercise.

## 4 WELL-POSEDNESS AND CONVERGENCE UNDER SHARP LIPSCHITZ BOUND

In this section we present the well-posedness of the backward SDE (2) and the convergence of discretization (4) based on sharp Lipchitz bound of score in Section 3. In Proposition 4.1, we review a well-posedness condition of a general SDE with additive noise where the drift term $F$ is only locally Lipschitz continuous. Combining it with Theorem 3.5, we present the well-posedness of (2) in

Theorem 4.3. Due to the limitation of global Hessian estimate, the convergence analysis is divided into the following two cases by $p_0$. The first case in Theorem 4.4 enjoys better complexity with respect to dimension ($N = \mathcal{O}(n \log^2 n)$) while has limitation in the final time $T$ in the forward process. The second case in Theorem 4.6 is valid globally in $T$ while achieving polynomial complexity ($N = \mathcal{O}(n^3 \log^2 n)$).

**Proposition 4.1.** *Given $T > 0$, suppose that $F = F(t, x) \in C([0, T] \times \mathbb{R}^n, \mathbb{R}^n)$ satisfies that $F$ is locally Lipschitz continuous in $x$ variable, i.e., for any $M > 0$, there exists a constant $L_M$ such that*

$$|F(t, x) - F(t, y)| \leq L_M |x - y| \quad \text{for } x, y \in B_M(0) \text{ and } t \in [0, T]$$

*and*

$$|F(t, x)| \leq C(|x| + 1). \quad \text{for } (t, x) \in [0, T] \times \mathbb{R}^n. \tag{17}$$

*for a positive constant C. For any $x_0 \in \mathbb{R}^n$, the following SDE has a unique solution*

$$dX_t = F(t, X_t)dt + dW_t, \quad t \in [0, T], \quad X_0 = x_0.$$

The proof is in Section C.7.

**Remark 4.2.** *For simplicity, uniform Lipschitz continuity of $F$ is often assumed to ensure the long-term existence of solutions for ODEs and SODEs. The above Theorem says that local Lipschitz continuity of $F$ plus the linear growth condition (17) would be sufficient, which is a special case of more general results ( see Theorem 2.4 and Theorem 3.1 in Chapter IV of Ikeda & Watanabe (2014) for instance). For reader's convenience, we presented the proof above for our special case.*

**Theorem 4.3.** *Let $g(x) = -\log p_0(x) - |x|^2/2$, be the non-Gaussian part of the log-density. We assume $g(x) \in C^{0,1}(\mathbb{R}^n)$ satisfy the Assumption 2.3 and $|\nabla g| \leq C(|x| + 1)$ for a positive constant $C$. [3] Then for any $T > 0$, the above (2) is well-posed.*

Proof is in Section C.7.

In what follows we denote the distribution of the discrete backward process (4) at generation time $T$ as $\hat{Q}_T$.

**Case I: $p_0$ is (near) log-concave**

**Theorem 4.4.** *Assume the following global Hessian bound of $p_0$,*

$$-L_1 I_n \preceq D^2 \log p_0(x) \preceq L_0 I_n.$$

*Let $\hat{q}_T$ be a distribution generated by the uniform discretization ($\delta = 0$) of the exponential integrator scheme (4), with an approximated score satisfying the Assumption 2.1. We also assume the $p_0$ has finite second order moment, namely $M_2 < \infty$ in (10). For $L_0 > 0$ and $T < -\log(1 - \frac{1}{L_0+1})$,*

$$\text{KL}(P_0 \| \hat{Q}_T) \lesssim (M_2 + n)e^{-T} + T\epsilon_0^2 + \frac{nT^2 C_T^2}{N}, \tag{18}$$

*where $C_T$ defined in (12) depends on $L_0$, $L_1$, $T$.*

*If $L_0 \leq 0$, namely $p_0$ is log-concave,*

$$\text{KL}(P_0 \| \hat{Q}_T) \lesssim (M_2 + n)e^{-T} + T\epsilon_0^2 + \frac{nT^2 C^2}{N}, \tag{19}$$

*where $C = \sup_{t \in [0,\infty)} (e^{-t} L_1 + (1 - e^{-t})) = \max\{L_1, 1\} < \infty$.*

**Proof:** We first apply Corollary 3.2 to attain global Hessian estimate in finite time. Then apply it to Proposition 4.1 for well-posedness and Proposition A.3 for convergence rate.

**Remark 4.5.** *(i) A near linear complexity bound, $N = \mathcal{O}(n \log^2 n)$, is then established by (19) under the log-concave distribution with $T = \mathcal{O}(\log n)$.*

*(ii) Note all complexity bounds by Proposition A.3 requires $T = \mathcal{O}(\log n)$ with second order moment $M_2 \lesssim n$. This implies the optimal bound with Lipschitz of score $L < \infty$, requires $N = \mathcal{O}(n \log^2 n)$.*

*(iii) Furthermore in the near log-concave case, we consider the regime with smallness of $L_0 = \mathcal{O}(\frac{1}{n})$. Therefore maximal time of estimate in (18), turns to $-\log(1 - \frac{1}{L_0+1}) = \mathcal{O}(\log(n))$. Then in (18) with $T = \mathcal{O}(\log(n))$, the first term is bounded and $C_T$ defined in (12) is independent with dimension $n$. The complexity bound in such case is also $\mathcal{O}(n \log^2 n)$.*

---

[3] Equivalently, $\log p_0$ is continuous differentiable, $|\nabla \log p_0| \leq C(|x| + 1)$ for a positive constant $C$.

**Case II: General smooth $p_0$**

**Theorem 4.6.** *Assume $||D^2g(x)||_2 \leq L$. Suppose that Assumption 2.4 holds and there exists $C_0 > 0$ such that, $\alpha_1 \leq C_0 n$ and $|\nabla g(x_0)| \leq C\sqrt{n}$. Let $\hat{q}_T$ be distribution generated by uniform discretization of the exponential integrator scheme (4), with an approximated score satisfying Assumption 2.1. We have,*

$$\mathrm{KL}(P_0\|\hat{Q}_T) \lesssim (M_2 + n)e^{-T} + T\epsilon_0^2 + \frac{CL^6Tn(n\log n)^2}{N}$$

**Proof:** It is a direct consequence of Theorem C.5 to estimate truncation error in Proposition A.1.

In addition to the above cases, we also consider non-smooth $p_0$ supported on compact manifold. Restricted by the estimate in (16), we switch to the early stopping technique, namely $\delta > 0$ in discretization. Due to the measure zero set (see in Section 3.3), the convergence bound is not yet optimal as shown in Section C.9.

**Remark 4.7.** *Bounds in Theorem 4.4 and Theorem 4.6 are consequences of our new Lipschitz estimate Theorem 3.1 and Theorem 3.5 with Proposition A.1 from Chen et al. (2023). An important feature of these new bounds is their uniformity in time (up to $T$, the mixing time of the forward process). Though the complexity in Theorem 4.6 is $\mathcal{O}((n\log n)^3)$ in dimension with $T = \mathcal{O}(\log n)$, slightly higher than $\mathcal{O}((n\log n)^2)$ in Theorem 2.5 of Chen et al. (2023), our assumption on time discretization $\{t_k\}_k$ during the sampling process (4) can be relaxed to uniform discretization. Hence our theory requires no prior knowledge of the data distribution $p_0$ and is more realistic.*

*We are also aware of two works which provide the linear in dimension ($\mathcal{O}(n)$) complexity bounds. Benton et al. (2024) utilizes a stochastic localization approach to attain the complexity bound [4] in the early stopping setting. A recent preprint Conforti et al. (2025) provides the linear bound (Theorem 1 of Conforti et al. (2025)) in a setting similar to our Theorem 4.6 with analysis of relative score process. In contrast to our work here, the approaches in Benton et al. (2024); Conforti et al. (2025) estimate the expected Lipschitz bound of the score under the backward process $\tilde{X}$ in (2), while our analysis is in the point-wise sense and hence applicable to the analysis of (approximated) score acting on the approximated backward process $\hat{x}$ in (4). Therefore our estimates readily apply to complexity and convergence bounds in the Wasserstein metric. A sketch of proof is presented in the appendix, Section C.10, which will be expanded in a future publication.*

## 5 CONCLUSION

The uniform spatial Lipschitz bound for $\nabla p(t, x)$, often assumed in the literature for various purposes, does not generally hold for all times, even with nice initial data (Theorem 3.4). Instead, we have established a local-in-space, global-in-time Lipschitz bound (Theorem 3.5) under mild assumptions, which can be employed to prove well-posedness and polynomial convergence of the KL divergence (Theorem 4.6)). As another potential application, a proof of sketch for obtaining convergence rate under Wasserstein distance is also provided in Appendix C.10 under similar assumptions.

For data on manifolds or non-smooth settings, we show that an $\mathcal{O}(1/t)$ Lipschitz bound exists almost everywhere (Theorem 3.8), suggesting that this type of bound can be reasonably assumed when studying convergence rates in the previous literature (for instance Theorem 3 in Bortoli (2022)).

**Limitation** Due to the limited knowledge of regularity factors of data distribution (e.g. optimal Lipschitz constants), our bound cannot provide implementable guidance on seeking the optimal schedule (which may require a separation of temporal regimes). Also in the manifold case, due to the complex geometries, as shown in the non-smooth section, our theories cannot provide a justifiable guidance of early stopping time. We will investigate these issues in a future study.

---

[4]in fact, the bound is $\mathcal{O}(n\log^2 n)$ due to Remark 4.5.

## ACKNOWLEDGMENTS

CM was partly supported by Alfred P. Sloan Research Fellowship, NSF CAREER grant DMS-2143668, Simons Fellowship. ZW was partly supported by NTU SUG-023162-00001, MOE AcRF Tier 1 Grant RG17/24. JX was partly supported by NSF grants DMS-2151235, DMS-2219904, DMS-2309520, a Qualcomm Gift Award. YY was partly supported by NSF grant DMS-2000191.

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

In the appendix below, we present detailed proof of theorems along with existing results used in the proof. Appendix A lists the KL-convergence theories of the diffusion model in Chen et al. (2023), which is applied when showing our convergence bound. Appendix B contains the theories in Alvarez et al. (1997) regarding the convexity result of second order differential equation of a very general kind. It is applied when showing the semi-convexity of the HJ equation, which is part of the global in space Hessian bound. Appendix C collects all the proofs and Appendix D discusses the broader impact of the manuscript.

## A  CONVERGENCE THEORIES

In this section, we list some numerical algorithms and convergence theories related to the numerical discretization of (2). They are due to Chen et al. (2023).

**Convergence in distribution**  The key ingredient of the convergence theory is the following result from the chain rule of KL divergence.

**Proposition A.1** (Prop C.3. of Chen et al. (2023))**.** *Given the score error estimation Assumption 2.1, the exponential integrator scheme* (4) *satisfies,*

$$\mathrm{KL}(P_\delta \| \hat{Q}_{T-\delta}) \lesssim \mathrm{KL}(P_T \| \gamma_n) + T\epsilon_0^2 + \sum_{k=1}^{N} \int_{t_{k-1}}^{t_k} E \| \nabla \log p(t, \tilde{X}_t) - \nabla \log p(t_k, x_{t_k}) \|^2 dt,$$

*where $\gamma_n$ is the Uniform Gaussian distribution*

The first term $\mathrm{KL}(P_T \| \gamma_n)$ in the estimate measures the distance between an the measure of OU process to its invariant measure. When the data has finite second order moment, it turns to 0 as $T \to \infty$.

**Proposition A.2** (Lem C.4 of Chen et al. (2023))**.** *With finite second order moment $E_{p_0}|X|^2 < \infty$, for $T > 1$,*

$$KL(P_T, \gamma_n) \leq (n + M_2)e^{-T}.$$

The third term relates to local truncation error that depends on the regularity of the forward process. Then Proposition A.1 can be further extended if the global Hessian estimate is available.

**Proposition A.3** (Theorem 2.1 of Chen et al. (2023))**.** *Given assumption of Proposition A.1 , $\nabla \log p_t$ is L-Lipschitz. For uniform discretization, the exponential integrator scheme* (4) *satisfies,*

$$\mathrm{KL}(P_0 \| \hat{Q}_T) \lesssim (M_2 + n)e^{-T} + T\epsilon_0^2 + \frac{nT^2L^2}{N}.$$

## B  SEMI-CONVEXITY OF SECOND ORDER DIFFERENTIAL EQUATION

Here we list some important theories used to construct the finite time log-convexity of the density of the forward process.

Given a function $w(t, x)$, its convex envelop $w^{**}(t, x)$ in $x$ is defined as

$$w^{**}(t, x) = \inf \left\{ \sum_{i=1}^{n+1} \lambda_i w(t, z_i) \big| \; x = \sum_{i=1}^{n+1} \lambda_i z_i, \; \sum_{i=1}^{n+1} \lambda_i = 1, \; \lambda_i \geq 0, z_i \in \mathbb{R}^n \right\}. \tag{20}$$

**Lemma B.1** (Prop 7 of Alvarez et al. (1997), Lemma 2 of Strömberg (2010))**.** *Let $w$ be a solution of,*

$$\partial_t w + F(t, x, \nabla w, D^2 w) = 0 \tag{21}$$

*The convex envelope $w^{**}$ of $w$ is a supersolution of* (21)*, under the following assumptions,*

1. *$F$ is elliptic in the sense $F(t, x, p, A) \geq F(t, x, p, \tilde{A})$ if $A \leq \tilde{A}$.*

2. *$(x, A) \in \mathbb{R}^n \times S_{++}^n \mapsto F(t, x, p, A^{-1})$ concave for all $t$ and $p$. Here $S_{++}^n$ is the set of all $n \times n$ positive definite matrices.*

    *3. w is coercive in the sense that*

$$\lim_{x \to \infty} \frac{w(t, x)}{|x|} = \infty, \tag{22}$$

    *uniformly in t.*

## C  PROOFS

In this section, we present proofs of the main theorems.

### C.1  PROOF OF THEOREM 3.1

For (1), it is equivalent to showing that

$$D^2 \bar{q}(t, x) \le M_1 I_n \quad \text{for all } (t, x) \in [0, 1] \times \mathbb{R}^n.$$

This actually follows immediate from (36). Here we will present a standard PDE approach that does not reply on the formula.

Let $\xi$ be a given unit vector. By taking derivatives of (6), we deduce that $v = q_{\xi\xi}$ satisfies

$$\partial_t v - \frac{1}{2} \Delta v + \nabla \bar{q} \cdot \nabla v = -|\nabla \bar{q}_\xi|^2 \le 0 \quad \text{on } (0, 1) \times \mathbb{R}^n.$$

Thanks to the standard maximum principle of parabolic equation, we have that

$$v(t, x) \le \sup_{x \in \mathbb{R}^n} v(x, 0) = \sup_{x \in \mathbb{R}^2} q_{\xi\xi} \le M_1.$$

The proof of (2) is more interesting. It is equivalent to showing that

$$D^2 \bar{q}(t, x) \ge -\frac{M_0}{1 - M_0 t} I_n. \tag{23}$$

Below we show a PDE approach that is base on modification of arguments in Strömberg (2010) for obtaining semiconcavity of solutions to the general viscous Hamilton-Jacobi equations.

Fix $\delta_1 > 0$. Note that by $D^2 g \succeq -M_0 I_n$,

$$g(x) - g(0) - \nabla g(0) \cdot x \ge -\frac{M_0}{2} |x|^2.$$

Hence there exists a constant $C_{\delta_1}$ such that

$$g(x) \ge -\left(\frac{M_0}{2} + \delta_1\right)|x|^2 - C_{\delta_1}.$$

Let $\alpha$ and $c$ be positive numbers satisfying that

$$\theta(0) = \alpha \tan(\alpha c) \ge M_{\delta_1} = M_0 + 4\delta_1$$

Consider the following construction,

$$w = \bar{q} + \theta(t)|x|^2/2 + n\Theta(t)/2, \tag{24}$$

where,

$$\theta(t) = \alpha \tan(\alpha c + \alpha t), \quad t < T^* = \frac{\pi}{2\alpha} - c \tag{25}$$

$$\Theta(t) = \int_0^t \theta(s)ds.$$

We notice that Eq.(25) implies, $\theta(0) = \alpha \tan(\alpha c)$ and $\theta' - \theta^2 = \alpha^2$. Then $w$ satisfies the following equation:

$$0 = \partial_t w + F(t, x, \nabla w, \nabla^2 w) := \partial_t w - \frac{1}{2}\Delta w + \frac{1}{2}|\nabla w|^2 - \theta(t)\nabla w \cdot x - \frac{\alpha^2}{2}|x|^2. \quad (26)$$

Now we consider the convex envelope (definition see (20)) of $w$, $w^{**}$ and aim to apply Lemma B.1 to show that $w^{**}$ is a supersolution of Eq.(26). After direct validation of the first two condition of Lemma B.1, it resorts to coercivity assumption (22). To this end, we construct a solution $\underline{q}$ of the equation ((6)) subjecting to $\underline{q}(0, x) \leq \bar{g}(x)$:

$$\underline{q}(t, x) = \theta_1(t)\frac{|x|^2}{2} + \Theta_1(t)\frac{n}{2} - C_{\delta_1}$$

where

$$\theta_1(t) = \frac{1}{t - \frac{1}{M_0 + 2\delta_1}}, \quad \Theta_1(t) = \int_0^t \theta_1(t)dt. \quad (27)$$

From Eq.(27), we know the construction holds for $t \in [0, \frac{1}{M_0 + 2\delta_1})$. By revisiting (25),

$$\sup_{\{\alpha \tan(\alpha c) \geq M_{\delta_1}\}} \left(\frac{\pi}{2\alpha} - c\right) = \lim_{\{\alpha \to 0^+, \ \alpha \tan(\alpha c) = M_{\delta_1}\}} \left(\frac{\pi}{2\alpha} - c\right) = \frac{1}{M_{\delta_1}} = \frac{1}{M + 4\delta_1}$$

and

$$\lim_{\{\alpha \to 0^+, \ \alpha \tan(\alpha c) = M_{\delta_1}\}} \theta(t) = \frac{M_{\delta_1}}{1 - M_{\delta_1}t}.$$

Then by choose suitable $\alpha$ and $c$, comparison principle of (6) which is equivalent to one of (8) says that

$$\bar{q}(t, x) \geq \underline{q}(t, x) \quad \text{for } t \in [0, \frac{1}{M_{\delta_1}})$$

and hence,

$$w \geq (\theta(t) + \theta_1(t))\frac{|x|^2}{2} + (\Theta(t) + \Theta_1(t))\frac{n}{2} \quad \text{for } t \in [0, \frac{1}{M_{\delta_1}}) \quad (28)$$

Now turning back to Eq.(28), we know $\theta(t) + \theta_1(t) \geq 2\delta_1 > 0$ uniform in any closed subinterval of $t \in [0, \frac{1}{M_{\delta_1}})$. This ensures the uniform coercivity requirement in Lemma B.1.

Summing up, by Lemma B.1, $w^{**}$ is a supersolution. On the other side, as convex envelope, $w^{**} \leq w$. Next, we want to utilize the comparison principle of (26) to show $w^{**} \geq w$ for all $t$, which is equivalent to Eq.(6) due to the construction (24). To do this, we only needs $w^{**}(0, X) \geq w(0, X)$, equivalently $w(0, X)$ is convex and it assured by $\theta(0) \geq M_{\delta_1} > M_0$.

Now we have $w = w^{**}$ for $x \in R^d$, $t \in [0, T^*]$, is convex. In particular, this implies that

$$D^2 \bar{q}(t, x) \geq -\theta(t)I_n \quad \text{for } (t, x) \in [0, T^*] \times \mathbb{R}^n.$$

Hence we derive that for any $T < \frac{1}{M_{\delta_1}}$,

$$\inf_{[0,T] \times \mathbb{R}^n} D^2 \bar{q}(t, x) \geq -\frac{M_{\delta_1}}{1 - M_{\delta_1}t}I_n. \quad (29)$$

Then (23) follows by sending $\delta_1 \to 0$.

Since the transform (7) only requires estimate of $\bar{q}$ for $t \in [0, 1]$, when $M_0 < 1$, (29) holds for any $T < 1$. Recalling transformation of $q$, the condition $M_0 < 1$ is equivalent to $-\log p$ is convex.

$\square$

### C.2 Proof of Theorem 3.4: Construction of Example of the Loss of Uniform Hessian Bound

Precisely speaking, we will construct a one dimensional ($n = 1$) example of $g(x) = G^2(x)$ for a smooth Lipschitz continuous function $G$ satisfying that $|G'(x)| \leq 1$, $|g''(x)| \leq 2 = M_0 = M_1$ and

$$\limsup_{|x| \to +\infty} |\bar{q}''(1/2, \ x)| = \infty.$$

For $M > 0$, let $g_M$ be the even function such that

$$g_M(x) = \begin{cases} 2M^2 - x^2, & 0 \leq x \leq M \\ (x - 2M)^2, & M \leq x \leq 2M \\ 0, & x \geq 2M. \end{cases}$$

Note that $|g_M''| \leq 2$ independent of $M$. Let $h_M(t, \ x)$ be the solution to the following heat equation

$$u_t - \frac{1}{2} \Delta u = 0 \quad \text{for } (t, x) \in (0, \infty) \times \mathbb{R} \tag{30}$$

subject to $h_M(0, x) = e^{-g_M(x)}$.

Since $h_M$ is even in $x$, $(h_M)_x(1/2, \ 0) = 0$. Hence, we have

$$(\log h_M)_{xx}(1/2, \ 0) = \frac{(h_M)_{xx}(1/2, \ 0)}{h_M(1/2, \ 0)}$$

$$= \frac{e^{-2M^2} \int_0^M (4y^2 + 2)\,dy + e^{-2M^2} \int_M^{2M} e^{-2(y-M)^2}(4(y - 2M)^2 - 2)\,dy}{e^{-2M^2} M + e^{-2M^2} \int_M^{2M} e^{-2(y-M)^2}\,dy + \int_{2M}^\infty e^{-y^2}\,dy}$$

$$:= \frac{A + B}{C + D + E}.$$

Clearly, for $M \geq 1$,

$$A + B = 4e^{-2M^2} \int_0^M y^2\,dy + 4e^{-2M^2} \int_M^{2M} e^{-2(y-M)^2}(y - 2M)^2\,dy > M^3 e^{-2M^2}.$$

and

$$C + D + E \leq 3M e^{-2M^2}.$$

Thus

$$(\log h_M)_{xx}(1/2, 0) > \frac{M^2}{3} \tag{31}$$

for all $M \geq 1$.

**Remark C.1.** *Note that $g_M$ is $C^{1,1}$, not smooth. However, the estimates above still hold for sufficiently fine mollifications of $g_M$, so we may assume without loss of generality that $g_M$ is smooth.*

Below we will choose a sequence $0 \leq x_1 \leq x_2 \leq ...$ such that the terms in $g := \sum_{k=1}^\infty g_k(x - x_k)$ have disjoint support, and such that

$$(\log \bar{p}_N)_{xx}(1/2, \ x_k) > \frac{k^2}{3}, \quad \text{for } k = 1, ..., N, \tag{32}$$

where $\bar{p}_N$ is the solution to (30) subject to $\bar{p}_N(0, x) = e^{-\sum_{k=1}^N g_k(x - x_k)}$. Note that $\bar{p}_N(0, x) = 1$ for $x \leq -2$.

Suppose we have managed to do this. Note that

$$\hat{p} \leq \bar{p}_N \leq 1$$

where $\hat{p}$ is the solution to (30) subject to $\hat{p}(0, x) = \chi_{[-3, -2]}$. Then interior derivative estimates for solutions of (30) imply that $(\log \bar{p}_N)_{xx}(1/2, \ \cdot)$ converge locally uniformly on $\mathbb{R}$ as $N \to \infty$ to $(\log \bar{p})_{xx}(1/2, \ \cdot, )$, where $q$ is the caloric function with initial data $e^{-g}$. Therefore, $(\log \bar{p})_{xx}(\cdot, 0)$ is bounded, but $(\log \bar{p})_{xx}(1/2, \ \cdot)$ is unbounded (its values at $x_k$ are at least $\frac{k^2}{3}$), as desired.

We now explain how to choose $x_k$. We will repeatedly use the fact that if $h$ is a bounded smooth function on $\mathbb{R}$ and $\tilde{h}$ is a compactly supported smooth function, then the caloric function with initial data $h + \tilde{h}(\cdot + S)$ converges in $C^2$ as $|S| \to \infty$ on compact subsets of $\{t > 0\}$ to the caloric function with initial data $h$.

First, we let $x_1 = 0$. Then (32) with $N = 1$ follows immediately from (31). Now suppose we have chosen $x_1 < ... < x_{M-1}$ such that the supports of $g_k(x - x_k)$, $1 \le k \le M - 1$ are disjoint and (32) holds for $N = M - 1$. Using the above-mentioned fact and (31), if we take $x_M$ sufficiently large, then (32) holds for $N = M$. Indeed, the inequality for $k < M$ follows immediately from the fact above, and the inequality for $k = M$ follows from the fact above and the inequality (31), after translating so that $x_M$ becomes 0. This completes the construction.

## C.3 PROOF OF THEOREM 3.5

Without loss of generality, we may assume $x_0 = 0$. It suffices to show the above for $|x| \ge 1$. For $|x| \le 1$, we can just replace $|x|$ in all the final bounds with 1.

First, we prove (13). Without loss of generality, let $g(0) = 0$. Then

$$g(z) \le \beta_{\frac{1}{2}} |z|^2 + \beta_2 |z| \quad \text{for } z \in \mathbb{R}^n.$$

Recall that

$$\bar{p}(t, x) = \frac{1}{(2\pi t)^{\frac{n}{2}}} \int_{\mathbb{R}^n} e^{-\frac{|x-y|^2}{2t}} h(y) \, dy$$

$$= \frac{1}{(\pi)^{\frac{n}{2}}} \int_{\mathbb{R}^n} e^{-|y|^2} h(x - \sqrt{2t}y) \, dy.$$

Then

$$-\nabla \bar{q} = \frac{\nabla \bar{p}}{\bar{p}} = \frac{1}{\bar{p}} \frac{1}{(\pi)^{\frac{n}{2}}} \int_{\mathbb{R}^n} e^{-|y|^2} h(x - \sqrt{2t}y) Dg \, dy$$

$$= \frac{1}{\bar{p}} \frac{1}{(\pi)^{\frac{n}{2}}} \int_{\mathbb{R}^n} e^{-|y|^2} h(x - \sqrt{2t}y) Dg \, dy \tag{33}$$

Since $ab \le a^2 + \frac{b^2}{2}$,

$$g(x - \sqrt{2t}y) \le \beta_1(|x|^2 + 2|y|^2) + \beta_2(|x| + \sqrt{2}|y|) \le 2\beta_1(|x|^2 + 2|y|^2) + \frac{\beta_2^2}{\beta_1},$$

we deduce that

$$\bar{p}(t, x) \ge e^{-\frac{\beta_2^2}{\beta_1}} \frac{e^{-2\beta_1|x|^2}}{(\pi)^{\frac{n}{2}}} \int_{\mathbb{R}^n} e^{-(1+4\beta_1)|y|^2} \, dy$$

$$= e^{-\frac{\beta_2^2}{\beta_1}} e^{-2\beta_1|x|^2} \left(\frac{1}{1+4\beta_1}\right)^{\frac{n}{2}}. \tag{34}$$

Then

$$|\nabla \bar{p}| = \frac{1}{(\pi)^{\frac{n}{2}}} \left| \int_{\mathbb{R}^n} e^{-|y|^2} Dh(x - \sqrt{2t}y) \, dy \right| = \frac{1}{(\pi)^{\frac{n}{2}}} \left| \int_{\mathbb{R}^n} e^{-|y|^2} h Dg(x - \sqrt{2t}y) \, dy \right|$$

$$\le \frac{1}{(\pi)^{\frac{n}{2}}} \int_{\mathbb{R}^n} (\beta_1|x| + \sqrt{2}\beta_1|y| + \beta_2) e^{-|y|^2} h(x - \sqrt{2t}y) \, dy$$

$$= (\beta_1|x| + \beta_2)\bar{p} + \frac{\sqrt{2}\beta_1}{(\pi)^{\frac{n}{2}}} \int_{\mathbb{R}^n} |y| e^{-|y|^2} h(x - \sqrt{2t}y) \, dy.$$

Let

$$\tilde{K} = \frac{2}{\sqrt{1 - \alpha_2}} \max\left\{ \frac{C_{n,m}}{|x|}, \, 4\sqrt{\beta_1} + 1 \right\},$$

where

$$C_{n,m} = 2\sqrt{(n+3)\log\left(\frac{2(1+4\beta_1)}{\sqrt{1-\alpha_2}}\right) + 4n\log n + \alpha_1 + J_m + \frac{\beta_2^2}{\beta_1}}.$$

Here for $m \in \mathbb{N}$, $J_m$ is the last positive integer such that $e^{r^2} \geq r^m$ for when $r \geq J_m$. In particular, $J_1 = J_2 = J_3 = 1$ and $C_n = C_{n,m}$ for $m = 1, 2, 3$.

**Claim:** If

$$K \geq K_0 = \max\left\{ \tilde{K}, \frac{6\alpha_2}{1 - \alpha_2} \right\},$$

then for $i = 1, 2, 3, ..., m$,

$$T_i(x) = \frac{1}{\bar{p}(t,x)} \frac{1}{(\pi)^{\frac{n}{2}}} \int_{y \in \mathbb{R}^n} |y|^i e^{-|y|^2} h(x - \sqrt{2t}y)\, dy \leq K^i |x|^i + 1 \leq 2K^i |x|^i. \tag{35}$$

Clearly, $T_i(0) = \mathbb{E}_{p(t,z)}(|z|^i)$.

Let us prove the claim. Note that

$$\bar{p}(t,x) T_i = \frac{1}{(\pi)^{\frac{n}{2}}} \int_{|y| \leq K|x|} |y|^i e^{-|y|^2} h(x - \sqrt{2t}y)\, dy + \frac{1}{(\pi)^{\frac{n}{2}}} \int_{|y| \geq K|x|} |y|^i e^{-|y|^2} h(x - \sqrt{2t}y)\, dy$$

$$\leq K^i |x|^i \bar{p}(t,x) + \underbrace{\frac{1}{(\pi)^{\frac{n}{2}}} \int_{|y|^i \geq K|x|} |y|^i e^{-|y|^2} h(x - \sqrt{2t}y)\, dy}_{I_i}.$$

Our goal is to show that $I_i \leq \bar{p}(t,x)$ when $K \geq K_0$. Since $g(z) \geq -\alpha_2 |z|^2 - \alpha_1$, $|y| \geq K|x|$ and $K > \frac{6\alpha_2}{1-\alpha_2}$,

$$g(x - \sqrt{2t}y) \geq -\frac{\alpha_2}{2} \left( |x| + \sqrt{2}|y| \right)^2 - \alpha_1$$

$$> -\alpha_2 |y|^2 \left( \frac{1}{K} + 1 \right)^2 - \alpha_1$$

$$> \alpha_2 |y|^2 \left( \frac{3}{K} + 1 \right) - \alpha_1$$

$$\geq -\frac{(1+\alpha_2)}{2} |y|^2 - \alpha_1$$

Then

$$I_i \leq \frac{e^{\alpha_1}}{(\sqrt{\pi})^n} \int_{|y| \geq K|x|} |y|^i e^{-\frac{(1-\alpha_2)|y|^2}{2}}\, dy.$$

For convenience, denote $K_1 = \frac{K\sqrt{1-\alpha_2}}{2}$. Then

$$\frac{e^{\alpha_1}}{(\sqrt{\pi})^n} \int_{|y| \geq K|x|} |y|^i e^{-\frac{(1-\alpha_2)|y|^2}{2}}\, dy = \frac{e^{\alpha_1}}{(\sqrt{\pi})^n} \left( \frac{2}{\sqrt{1-\alpha_2}} \right)^{n+i} \int_{|z| \geq K_1|x|} |z|^i e^{-2|z|^2}\, dz$$

$$\leq \frac{e^{\alpha_1}}{(\sqrt{\pi})^n} \left( \frac{2}{\sqrt{1-\alpha_2}} \right)^{n+3} \int_{|z| \geq K_1|x|} e^{-|z|^2}\, dz.$$

The last inequality is due to $e^{r^2} \geq r^m$ if $r \geq J_m$ and $K_1|x| \geq J_m$.

Note

$$\left( \int_{-r}^{r} e^{-t^2}\, dt \right)^2 \geq \int_{\{w \in \mathbb{R}^2 \mid |y| \leq r\}} e^{-|w|^2}\, dw = \pi \left( 1 - e^{-r^2} \right).$$

Combining with $(1-t)^n \geq 1 - nt$ for $t \in [0,1]$, we deduce

$$\int_{Q_r} e^{-|y|^2}\, dy = \left( \int_{-r}^{r} e^{-t^2}\, dt \right)^n \geq (\sqrt{\pi})^n \left( 1 - e^{-r^2} \right)^{\frac{n}{2}} \geq (\sqrt{\pi})^n \left( 1 - \frac{ne^{-r^2}}{2} \right)$$

Here $Q_r = \{y = (y_1, y_2, ..., y_n) \mid |y_i| \leq r \text{ for } i = 1, 2, ..., n\}$. Accordingly, for $r \geq 1$,

$$\frac{1}{(\sqrt{\pi})^n} \int_{|y| \geq r} e^{-|y|^2}\, dy \leq \frac{1}{(\sqrt{\pi})^n} \int_{\mathbb{R}^n \setminus Q_r} e^{-|y|^2}\, dy + \frac{1}{(\sqrt{\pi})^n} \int_{Q_r \setminus \{|y| \leq r\}} e^{-|y|^2}\, dy$$

$$\leq \frac{ne^{-r^2}}{2} + \frac{1}{(\sqrt{\pi})^n} e^{-r^2} (2r)^n < 2^n e^{-r^2} r^n$$

This implies that

$$\frac{1}{(\sqrt{\pi})^n} \int_{|z|<K_1|x|} e^{-|z|^2} \le 2^n e^{-K_1^2|x|^2} (K_1|x|)^n.$$

Since $K_1 \ge \max\{3\sqrt{\beta_1}, \frac{2\sqrt{n\log n}}{|x|}\}$,

$$e^{-2\beta_1|x|^2} K_1^n |x|^n e^{-K_1^2|x|^2} \le e^{-2\beta_1|x|^2} e^{-\frac{K_1^2|x|^2}{2}} \le e^{-\frac{K_1^2|x|^2}{4}}$$

Combining with

$$\frac{K_1^2|x|^2}{4} \ge \frac{\tilde{K}^2|x|^2}{4} \ge (n+3)\log\left(\frac{1+4\beta_1}{\sqrt{1-\alpha_2}}\right) + 16n\log n + \alpha_1 + 1 + \frac{\beta_2^2}{\beta_1},$$

we obtain

$$e^{-2\beta_1|x|^2} I_i \le 2^n e^{\alpha_1} \left(\frac{2}{\sqrt{1-\alpha_2}}\right)^{n+3} e^{-\frac{K_1^2|x|^2}{4}} \le e^{-\frac{\beta_2^2}{\beta_1}} (1+4\beta_1)^{-\frac{n}{2}}.$$

By (34),

$$I_i \le \bar{p}(t,x).$$

Hence (35) holds. As an immediate conclusion, we have that

$$|\nabla \bar{q}| = \frac{|\nabla \bar{p}|}{\bar{p}} \le \beta_1|x| + \beta_2 + 2\beta_1\sqrt{2}K|x| < 3\beta_1 K|x| + \beta_2.$$

Secondly, to prove (14) and (15), we first assume that $g(0) = 0$ and $\nabla g(0) = 0$, which will be removed at the end. Then

$$|\nabla g(x)| \le L|x|.$$

Next we first verify (14). Note that

$$D^2\bar{q} = A - B + \nabla\bar{q} \otimes \nabla\bar{q} \tag{36}$$

Here

$$A = \frac{1}{\bar{p}} \frac{1}{(\pi)^{\frac{n}{2}}} \int_{\mathbb{R}^n} e^{-|y|^2} h D^2 g(x - \sqrt{2t}y)\, dy$$

and

$$B = \frac{1}{\bar{p}} \frac{1}{(\pi)^{\frac{n}{2}}} \int_{\mathbb{R}^n} e^{-|y|^2} h \nabla g \otimes \nabla g(x - \sqrt{2t}y)\, dy.$$

Here $(u \otimes u)_{ij} = u_i u_j$ is the outer product. By Cauchy inequality, $B \ge \nabla\bar{q} \otimes \nabla\bar{q}$. Hence

$$D^2\bar{q} \le A \le LI_n.$$

For the other direction,

$$D^2\bar{q} \ge A - B \ge -LI_n - B.$$

So we just need to estimate the term B. Note that

$$\left\| \frac{1}{(\pi)^{\frac{n}{2}}} \int_{\mathbb{R}^n} e^{-|y|^2} h \nabla g \otimes \nabla g(x - \sqrt{2t}y)\, dy \right\|_2 \le \frac{L^2}{(\pi)^{\frac{n}{2}}} \int_{\mathbb{R}^n} (|x| + \sqrt{2}|y|)^2 e^{-|y|^2} h\, dy$$

$$\le \frac{2L^2}{(\pi)^{\frac{n}{2}}} \int_{\mathbb{R}^n} (|x|^2 + 2|y|^2) e^{-|y|^2} h\, dy$$

$$\le 2L^2 \left( |x|^2 \bar{p}(t,x) + \frac{2}{\pi^{\frac{n}{2}}} \int_{\mathbb{R}^n} |y|^2 e^{-|y|^2} h\, dy \right).$$

Thanks to (35) for $\beta_1 = L, \beta_2 = 0$ and $K = K_0$, we have that

$$\|B\|_2 \le 2L^2(|x|^2 + 4K_0^2|x|^2) < 10L^2 K_0^2|x|^2.$$

Hence

$$\|D^2\bar{q}\|_2 \le L + \|B\|_2 < L + 10L^2 K_0^2|x|^2$$

Then let us verify (15). Note that

$$\partial_t \nabla \bar{q}(t,x) = C + D - \partial_t \bar{q} \nabla \bar{q}.$$

with

$$|C| = \left| \frac{1}{\sqrt{2t}} \frac{1}{\bar{p}} \frac{1}{(\pi)^{\frac{n}{2}}} \int_{\mathbb{R}^n} e^{-|y|^2} h D^2 g(x - \sqrt{2t}y) \cdot y \, dy \right| \le \frac{L}{\sqrt{2t}} \frac{1}{\bar{p}} \frac{1}{(\pi)^{\frac{n}{2}}} \int_{\mathbb{R}^n} |y| e^{-|y|^2} h \, dy$$

$$\le \frac{2L}{\sqrt{2t}} K|x| < \frac{2L}{\sqrt{t}} K|x|.$$

Also,

$$|D| = \left| \frac{1}{\sqrt{2t}} \frac{1}{\bar{p}} \frac{1}{(\pi)^{\frac{n}{2}}} \int_{\mathbb{R}^n} e^{-|y|^2} h \nabla g \otimes \nabla g(x - \sqrt{2t}y) \cdot y \, dy \right|$$

$$\le \frac{1}{\sqrt{2t}} \frac{2L^2}{(\pi)^{\frac{n}{2}}} \int_{\mathbb{R}^n} (|x|^2 |y| + 2|y|^3) e^{-|y|^2} h \, dy$$

$$< \frac{1}{\sqrt{t}} \frac{2L^2 |x|^2}{(\pi)^{\frac{n}{2}}} \int_{\mathbb{R}^n} |y| e^{-|y|^2} h \, dy + \frac{1}{\sqrt{t}} \frac{4L^2}{(\pi)^{\frac{n}{2}}} \int_{\mathbb{R}^n} |y|^3 e^{-|y|^2} h \, dy$$

$$\le \frac{2L^2}{\sqrt{t}} (2K|x|^3 + 4K^3 |x|^3)$$

$$< \frac{12L^2}{\sqrt{t}} K^3 |x|^3.$$

Finally,

$$|\partial_t \bar{q}| = \left| \frac{1}{\sqrt{2t}} \frac{1}{\bar{p}} \frac{1}{(\pi)^{\frac{n}{2}}} \int_{\mathbb{R}^n} e^{-|y|^2} h \nabla g(x - \sqrt{2t}y) \cdot y \, dy \right|$$

$$\le \frac{L}{\sqrt{2t}} \frac{1}{\bar{p}} \frac{1}{(\pi)^{\frac{n}{2}}} \int_{\mathbb{R}^n} (|x||y| + \sqrt{2}|y|^2) e^{-|y|^2} h \, dy$$

$$\le \frac{L}{\sqrt{2t}} (2K|x|^2 + 2\sqrt{2} K^2 |x|^2) \le \frac{6L}{\sqrt{t}} K^2 |x|^2.$$

Also,

$$|\nabla \bar{q}| \le 2L(1 + \sqrt{2}) K|x| < 6LK|x|.$$

Hence

$$|\partial_t \bar{q} \nabla q| \le \frac{36L^2}{\sqrt{t}} K^3 |x|^3.$$

So

$$|\partial_t \nabla \bar{q}(t,x)| \le \frac{2L}{\sqrt{t}} K|x| + \frac{1}{\sqrt{t}} 12L^2 K^3 |x|^3 + \frac{36L^2}{\sqrt{t}} K^3 |x|^3$$

$$\le \frac{50L^2}{\sqrt{t}} K^3 |x|^3.$$

At last, for a general $g$, we consider

$$g_0(x) = g(x) - g(0) - \nabla g(0) \cdot x.$$

and $\bar{q}_0(t,x)$ be the corresponding score function. Then we have

$$g_0(0) = 0, \quad \nabla g_0(0) = 0 \quad \text{and} \quad \bar{q}_0(t,x) = \bar{q}(t, x + \nabla g(0)).$$

Hence (14) and (15) hold for general cases. □

**Remark C.2.** *In the proof of* (15)*, we may also bound $\partial_t \bar{q}$ use the equation*

$$\partial_t \bar{q} = \frac{1}{2} (\Delta \bar{q} - |\nabla \bar{q}|^2)$$

*together with bounds for $\Delta \bar{q}$ and $|\nabla \bar{q}|^2$. Note $\nabla g(x - \sqrt{t}y) = \nabla g(x) + r_t$ for $|r_t| \le L\sqrt{t}|y|$. Then term $D$ in the proof*

$$D = \frac{1}{\sqrt{2t}} \frac{1}{\bar{p}} \frac{1}{(\pi)^{\frac{n}{2}}} \int_{\mathbb{R}^n} e^{-|y|^2} h \nabla g \otimes \nabla g(x - \sqrt{2t}y) \cdot y \, dy$$
$$= \partial_t \bar{q} \nabla g(x) + O(L^2 K^3 |x|^3).$$

*This will lead to a bound of $\partial_t \nabla \bar{q}$ by replacing $\frac{L^2}{\sqrt{t}}$ in (15) by $O(nL^3)$. The details are left to interested readers as an exercise.*

Below are other simple situations where we can obtain global uniform bound of the Hessian, which follows immediately from (33) and (36).

**Theorem C.3.** *Let $L_1$ and $L_2$ be two positive constants such that*

$$|\nabla g| \leq L_1 \quad \text{and} \quad ||D^2 g||_2 \leq L_2.$$

*Then*

$$|\nabla \bar{q}(t,x)| \leq L_1 \quad \text{and} \quad -(L_2 + L_1^2)I_n \leq D^2 q(t,x) \leq L_2 I_n$$

### C.4 PROOF OF (16)

We first prove the blow up bound of the gradient (1) in (16). Note that $D\bar{q} = -\frac{D\bar{p}}{\bar{p}}$ and

$$
\begin{aligned}
\bar{p}(t,x) &= \frac{1}{(2\pi t)^{\frac{n}{2}}} \int_{\mathbb{R}^n} e^{-\frac{|x-y|^2}{2t}} d\pi_0(y) \\
&= \frac{1}{(2\pi t)^{\frac{n}{2}}} \int_{D_0} e^{-\frac{|x-y|^2}{2t}} d\pi_0(y).
\end{aligned}
$$

Then

$$\nabla \bar{p}(t,x) = -\frac{1}{(2\pi t)^{\frac{n}{2}}} \int_{D_0} \frac{(x-y)}{t} e^{-\frac{|x-y|^2}{2t}} d\pi_0(y).$$

Since $D_0 \subset \overline{B}_M(0)$

$$|\nabla \bar{p}(t,x)| \leq \frac{|x| + M}{t} \bar{p}(t,x).$$

Next we prove the blow up bound of Hessian (2) in (16). Note that

$$-D^2 \bar{q}(t,x) = \frac{D^2 \bar{p}}{\bar{p}} - \frac{\nabla \bar{p} \otimes \nabla \bar{p}}{\bar{p}^2} = -\frac{\delta_{ij}}{t} + \frac{1}{t^2} \frac{A-B}{\hat{p}^2(t,x)}.$$

Here $\hat{p}(t,x) = \int_{D_0} e^{-\frac{|x-y|^2}{2t}} d\pi_0(y)$,

$$A_{ij} = \hat{p}(t,x) \int_{D_0} (x_i - y_i)(x_j - y_j) e^{\frac{-|x-y|^2}{2t}} d\pi_0(y)$$

and

$$
\begin{aligned}
B_{ij} &= \int_{D_0} (x_i - y_i) e^{\frac{-|x-y|^2}{2t}} d\pi_0(y) \cdot \int_{D_0} (x_j - y_j) e^{\frac{-|x-y|^2}{2t}} d\pi_0(y) \\
&= \hat{p}^2(t,x) x_i x_j - \hat{p}(t,x) \int_{D_0} (x_i y_j + x_j y_i) e^{\frac{-|x-y|^2}{2t}} d\pi_0(y) + \\
&\quad + \int_{D_0} y_i e^{\frac{-|x-y|^2}{2t}} d\pi_0(y) \cdot \int_{D_0} y_j e^{\frac{-|x-y|^2}{2t}} d\pi_0(y)
\end{aligned}
$$

Hence $(A-B)_{ij}$

$$\hat{p}(t,x) \int_{D_0} y_i y_j e^{\frac{-|x-y|^2}{2t}} d\pi_0(y) - \int_{D_0} y_i e^{\frac{-|x-y|^2}{2t}} d\pi_0(y) \cdot \int_{D_0} y_j e^{\frac{-|x-y|^2}{2t}} d\pi_0(y) \quad (37)$$

So it is easy to see that

$$||A - B||_2 \leq M^2 \hat{p}^2(t,x).$$

Thus (16) holds. □

### C.5 PROOF OF THEOREM 3.8

Since $\pi_0$ is comparable to the uniform distribution, there exists a constant $C$ such that for any measurable subset $U \subset D_0$

$$\frac{1}{C} \mathcal{H}_d(U) \leq \pi_0(S) \leq C \mathcal{H}_d(U)$$

Here $\mathcal{H}_d(\cdot)$ is the $d$-dimensional Hausdorff measure. Hereafter, we write

    (i) $\partial D_0$: the $d-1$ dimensional boundary of $D_0$. Moreover, for $y \in D_0$;

    (ii) $T_y(D_0) \subset \mathbb{R}^n$: the $d$-dimensional tangent space of $D_0$ at $y$;

(iii) $N_y(D_0) \subset \mathbb{R}^n$: the $n - d$ dimensional orthogonal complement of $T_y(D_0)$;

(iv) $T'_y(D_0) \subset \mathbb{R}^n$: the $d-1$-dimensional tangent space of $\partial D_0$ at $y \in \partial D_0$. Note that $T'_y(D_0)$ is a subspace of $T_y(D_0)$;

(v) $N'_y(D_0) \subset \mathbb{R}^n$: the $n + 1 - d$ dimensional orthogonal complement of $T'_y(D_0)$. $N_y(D_0)$ is a subspace of $N'_y(D_0)$

Let
$$S = \{x \in \mathbb{R}^n | \text{ there exists a unique } y_x \in D_0 \text{ such that } |x - y_x| = d(x, D_0)\}.$$
Then $\mathbb{R}^n \backslash S$ has zero measure since $d(x, D_0)$ is differentiable almost everywhere.

Write
$$S_1 = \{x \in S | \; y_x \in D_0 \backslash \partial D_0\} \quad \text{and} \quad S_2 = \{x \in S | \; y_x \in \partial D_0\},$$
$$W_1 = \left\{ x \in S_1 | \; \liminf_{y \in D_0 \to y_x} \frac{|x - y|^2 - |x - y_x|^2}{|y - y_x|^2} > 0 \right\},$$
$$W_2 = \left\{ x \in S_2 | \; x - y_x \in N_{y_x}(D_0) \quad \text{and} \quad \liminf_{y \in D_0 \to y_x} \frac{|x - y|^2 - |x - y_x|^2}{|y - y_x|^2} > 0 \right\},$$
and
$$W_3 = \left\{ x \in S_2 | \; x - y_x \notin N_{y_x}(D_0) \quad \text{and} \quad \liminf_{y \in \partial D_0 \to y_x} \frac{|x - y|^2 - |x - y_x|^2}{|y - y_x|^2} > 0 \right\}.$$

Note that
$$\text{if } y \in D_0 \backslash \partial D_0, \quad \text{then } x - y_x \in N_{y_x}(D),$$
$$\text{if } y \in \partial D_0, \quad \text{then } x - y_x \in N'_{y_x}(D).$$

**Step 1:** We show that $S \backslash \left( \cup_{i=1}^{3} W_i \right) = (S_1 \backslash W_1) \cup (S_2 \backslash W_2) \cup (S_2 \backslash W_3)$ has zero measure. We will prove this for $S_1 \backslash W_1$, The proofs for the other two are similar. Apparently, if $x \in S$, then for all $t \in (0, 1)$, $y_{\tilde{X}_t} = y_x$ and $\tilde{X}_t \in W$ for $\tilde{X}_t = y_x + t(x - y_x)$. Also,

For $y \in D_0 \backslash \partial D_0$, write
$$\Gamma_y = \{x \in S_1 \backslash W_1 | \; y_x = y\}.$$
By compactness argument, it is easy to show that for given $x \in S$ and $r > 0$, there exists a $r_x > 0$, such that $y_{\tilde{x}} \in B_r(y_x)$ for any $\tilde{x} \in B_{r_x}(x) \cap S$. Hence to prove that $S_1 \backslash W$ has zero measure, it suffices to show that for any $y_0 \in D_0$, if $\Gamma_{y_0}$ is not empty, then there exists $r_0 > 0$ such that
$$\cup_{y \in B_{r_0}(y_0) \cap D_0} \Gamma_y \subset \{y + t(y, v) | \; y \in B_{r_0}(y_0), \; v \in N_y(D_0) \text{ and } |v| = 1\} \qquad (38)$$
for a locally Lipschitz continuous function
$$t(y, v) : B_{r_0}(y_0) \times N_y(D_0) \to (0, \infty).$$

By suitable translation and rotation, we may assume $y_0 = 0$ and in a neighbourhood $V$ of $0$,
$$D_0 \cap V = V \cap \{(y', F(y')) | \; y' \in \mathbb{R}^d\}, \qquad (39)$$
where $F = (F^{(d+1)}, F^{(d+2)}, ..., F^{(n)}) : \mathbb{R}^d \to \mathbb{R}^{n-d}$ is smooth map satisfying $\nabla F(0) = 0$ and $F(0) = 0$. Choose $x \in \Gamma_{y_0}$. Let $v = \frac{x}{|x|} \in N_y(D_0)$ and $t > 0$, let
$$H_{tv}(y') = |tv - (y', F(y'))|^2.$$

Since $x \in W_1$, $D^2 H_{tv}(0)$ can not be a positive definite matrix for $t = |x|$. Meanwhile, for $0 < t < |x|$, $y_{tx} = y_0$ and $tx \in W_1$, which implies that $D^2 H_{tv}(0)$ is positive definite for $t \in [0, |x|)$. Hence $t = |x|$ is the first moment such that $D^2_{tv} H(0)$ has a zero eigenvalue. Which is equivalent to

the largest eigenvalue of the $d \times d$ matrix $\sum_{k=d+1}^{n} t v_k F^{(k)}_{y'_i y'_j}(0)$ is 1.

Therefore, for $y \in B_r(y_0)$ and $v \in T_y(D_0)$ with $|v| = 1$, if the largest eigenvalue $\lambda(y, v)$ of the matrix $\sum_{k=d+1}^{n} v_k F^{(k)}_{y'_i y'_j}(y)$ is positive, we set
$$t(y, v) = \frac{1}{\lambda(y, v)}.$$

Then (38) holds.

**Step 2:** We will verify that if $x \in \cup_{i=1}^3 W_i$, then

$$|D^2 \bar{q}(t,x)| \leq \frac{C_x}{t} \quad \text{for all } t \in (0,1].$$

Case 1: Assume that $x \in W_1$. Without loss of generality, we may assume $y_x = 0$ and use the representation as (39). Choose $r > 0$ such that

(i)$V_r = \{(y', F(y'))| \, |y'| < r\} \subset D_0$;

(ii) Then there exits $\alpha_x, \beta_x > 0$ such that for $y \in V_r$,

$$\alpha_x |y - y_x|^2 \geq |x - y|^2 - |x - y_x|^2 \geq \beta_x |y - y_x|^2 \quad \text{for } y \in D_0. \tag{40}$$

For $k \geq 1$ and $k\sqrt{t} \leq r$, write

$$V_{t,k} = \left\{ (y', F(y'))| \, |y'| < k\sqrt{t} \right\}$$

Thanks to the left upper bound in (40),

$$\int_{D_0} e^{\frac{-|x-y|^2}{2t}} \, d\pi_0 \geq \int_{V_{t,1}} e^{\frac{-|x-y|^2}{2t}} \, d\pi_0 \geq O\left(t^{\frac{d}{2}} e^{-\frac{|x-y_x|^2}{2t}}\right).$$

Recall that $y_x = 0$. To see the dependence on $y_x$, we keep $y_x$ in the computations below instead of replacing it by $0$. Note

$$\int_{D_0} |y - y_x|^2 e^{\frac{-|x-y|^2}{2t}} \, d\pi_0 \leq \int_{V_r} |y - y_x|^2 e^{\frac{-|x-y|^2}{2t}} \, d\pi_0 + Ce^{-\frac{|x-y_x|^2 + \delta_r}{2t}}$$

for some $\delta_r > 0$.

Also,

$$\int_{V_r} |y - y_x|^2 e^{\frac{-|x-y|^2}{2t}} \, d\pi_0 = \sum_{k=0}^{\infty} \int_{\{y \in V_{t,k+1} \setminus V_{t,k}\}} |y - y_x|^2 e^{\frac{-|x-y|^2}{2t}} \, d\pi_0$$
$$\leq Ct \cdot t^{\frac{d}{2}} e^{-\frac{|x-y_x|^2}{2t}} \sum_{k=1}^{\infty} (k+1)^2 e^{-k} = O\left(t^{\frac{d}{2}} e^{-\frac{|x-y_x|^2}{2t}}\right) t.$$

Hence

$$\frac{\int_{D_0} |y - y_x|^2 e^{\frac{-|x-y|^2}{2t}} \, d\pi_0(y)}{\hat{p}} \leq Ct \tag{41}$$

Recall that

$$\hat{p}(t,x) = \int_{D_0} e^{\frac{-|x-y|^2}{2t}} \, d\pi_0(y) = (\sqrt{2\pi t})^n \bar{p}(t,x).$$

Then for $1 \leq i, j \leq n$ and $y_x = (a_1, a_2, ..., a_n)$,

$$\left| \frac{\int_{D_0} y_i y_j e^{\frac{-|x-y|^2}{2t}} d\pi_0(y)}{\hat{p}(t,x)} - \frac{\int_{D_0} y_i e^{\frac{-|x-y|^2}{2t}} d\pi_0(y)}{\hat{p}(t,x)} \cdot \frac{\int_{D_0} y_j e^{\frac{-|x-y|^2}{2t}} d\pi_0(y)}{\hat{p}(t,x)} \right| =$$

$$\left| \frac{\int_{D_0} (y_i - a_i)(y_j - a_j) e^{\frac{-|x-y|^2}{2t}} d\pi_0(y)}{\hat{p}(t,x)} - \frac{\int_{D_0} (y_i - a_i) e^{\frac{-|x-y|^2}{2t}} d\pi_0(y)}{\hat{p}(t,x)} \cdot \frac{\int_{D_0} (y_j - a_j) e^{\frac{-|x-y|^2}{2t}} d\pi_0(y)}{\hat{p}(t,x)} \right|$$

$$\leq Ct.$$

The last equality follows from (41) and the Cauchy inequality. Therefore, (37) leads to

$$|D^2 \bar{q}(t,x)| \leq \frac{C_x}{t}.$$

Case 2: $x \in W_2$. The proof is similar to Case 1.

Case 3: $x \in W_3$. By suitable translation and rotation, we may assume $y_x = 0$ and in a neighborhood of $0 \in \mathbb{R}^d$,

$$D_0 \cap V = \tilde{V}_r = \{(y', F(y'))| \, y' = (y'_1, ..., y'_d) \in \Omega_{f,r}\},$$

where $F : \mathbb{R}^d \to \mathbb{R}^{n-d}$ is smooth map satisfying that $\nabla F(0) = 0$. Also,

$$\Omega_{f,r} = \{z = (z', z_d) | \; z' = (z_1, z_2, .., z_{d-1}) \in \mathbb{R}^{d-1}, \; |z'| < r; \; z_d \geq f(z_1, z_2, .., z_{d-1})\}.$$

for a smooth function $f : \mathbb{R}^{d-1} \to \mathbb{R}$ subject to $\nabla f(0) = 0$. Then

$$T_0(D_0) = \{(v, 0, ..., 0) \in \mathbb{R}^n \mid v \in \mathbb{R}^d\}$$

and the $d - 1$ dimensional tangent plane to $\partial D_0$ at $y_x = 0$ is

$$\partial T_0(D_0) = \{(v', 0, 0, ..., 0) \in \mathbb{R}^n \mid v' \in \mathbb{R}^{d-1}\}.$$

Thus

$$x = x - y_x = (\underbrace{0, ...0}_{d-1}, \theta_x, z_x)$$

for some $\theta_x > 0$ and $z_x \in \mathbb{R}^{n-d}$. To see the dependence on $y_x$, as in Case 1, we keep $y_x$ in the computations below instead of replacing it by $0$.

Then for $y'' \in \mathbb{R}^{d-1}$ and $y = (y'', y_d, F(y'', y_d)) \in D_0$,

$$|x - y|^2 - |x - y_x|^2 = 2\theta_x(y_d - f(y')) + O(|y - y_x|^2). \tag{42}$$

Write

$$H(y) = |x - y|^2 - |x - y_x|^2 = H(y'', y_d, F(y'', y_d)).$$

Since $x \in W_3$,

$$H(y'', f(y''), F(y'', f(y''))) \geq \delta_x |y''|^2.$$

Therefore, there exists $r > 0$ and $M > 0$ such that

$$|x - y|^2 - |x - y_x|^2 \geq \frac{\theta_x}{M}(y_d - f(y')) + \delta_x |y - y_x|^2 \quad \text{for all } y \in \tilde{V}_r.$$

Write

$$R_{t,k} = \{(y', y_d, F(y', y_d)) \in \Omega | \; |y'| \leq k\sqrt{t} \quad \text{and} \quad 0 \leq y_d - f(y') \leq kt\}.$$

Thanks to (42),

$$\hat{p}(t, x) = \int_{D_0} e^{\frac{-|x-y|^2}{2t}} \, d\pi_0 \geq \int_{D_0 \cap R_{t,1}} e^{\frac{-|x-y|^2}{2t}} \, d\pi_0 \geq O\left(t^{d+\frac{1}{2}} e^{-\frac{|x-y_x|^2}{2t}}\right).$$

Note that

$$\int_{D_0} |y - y_x|^2 e^{\frac{-|x-y|^2}{2t}} \, d\pi_0 \leq \int_{\tilde{V}_r} |y - y_x|^2 e^{\frac{-|x-y|^2}{2t}} \, d\pi_0 + Ce^{-\frac{|x-y_x|^2 + \delta_r}{2t}}.$$

Also,

$$\int_{\tilde{V}_r} |y - y_x|^2 e^{\frac{-|x-y|^2}{2t}} \, d\pi_0 = \sum_{k=0}^{\infty} \int_{D_0 \cap (R_{t,k+1} \setminus R_{t,k})} |y - y_x|^2 e^{\frac{-|x-y|^2}{2t}} \, d\pi_0$$
$$\leq Ct \cdot t^{d+\frac{1}{2}} e^{-\frac{|x-y_x|^2}{2t}} \sum_{k=1}^{\infty} (k+1)^2 e^{-k} = tO\left(t^{d+\frac{1}{2}} e^{-\frac{|x-y_x|^2}{2t}}\right).$$

Hence

$$\frac{\int_{D_0} |y - \bar{y}_x|^2 e^{\frac{-|x-y|^2}{2t}} \, d\pi_0(y)}{\hat{p}} \leq Ct$$

Then by the same argument in the end of Case 1, we deduce that

$$\|D^2 \bar{q}(t, x)\|_2 \leq \frac{C_x}{t}.$$

Finally, if $D_0$ is convex, then it is clear that $S = \mathbb{R}^n$ and $W_1 = S_1$ and $W_2 \cup W_3 = S_2$. Hence the $O(\frac{1}{t})$ bound holds for all $x \in \mathbb{R}^n$. □

## C.6 Proof of Example 3.9

Proof: For given $x \in \mathbb{R}^2$, denote by $\bar{y}_t$ the weighted center of mass:

$$\bar{y}_t = \frac{\int_{D_0} y e^{\frac{-|x-y|^2}{2t}} d\pi_0(y)}{\hat{p}}$$

Note that as $t \to 0$, the measure $\frac{e^{\frac{-|x-y|^2}{2t}} d\pi_0(y)}{\hat{p}}$ will concentrate on $\{y \in D_0 | \ |y - x| = d(x, D_0)\}$. Thus,

$$\lim_{t \to 0} d(\bar{y}_t, \Gamma_x) = 0,$$

where $\Gamma_x$ is the convex hull of $\{y \in D_0 | \ |y - x| = d(x, D_0)\}$. According to the computation in the proof of (16), we have that

$$-\Delta \bar{q} = -\frac{n}{t} + \frac{\hat{p} \int_{D_0} |y|^2 e^{\frac{-|x-y|^2}{2t}} d\pi_0(y) - \left|\int_{D_0} y e^{\frac{-|x-y|^2}{2t}} d\pi_0(y)\right|^2}{t^2 \hat{p}^2}$$

$$= -\Delta \bar{q} = -\frac{n}{t} + \frac{\int_{D_0} |y - \bar{y}_t|^2 e^{\frac{-|x-y|^2}{2t}} d\pi_0(y)}{t^2 \hat{p}},$$

where the second term is like a variance. If $x = (\theta, 0)$ for some $\theta > 0$, there are two points $y_1$ and $y_2$ such that

$$|x - y_1| = |x - y_2| = d(x, D_0).$$

Due to the symmetry, we must have that

$$\frac{e^{\frac{-|x-y|^2}{2t}} d\pi_0(y)}{\hat{p}} \to \frac{1}{2}\delta_{y_1} + \frac{1}{2}\delta_{y_2} \quad \text{and} \quad \lim_{t \to +\infty} y_t = \frac{y_1 + y_2}{2}.$$

Accordingly,

$$\lim_{t \to 0} \frac{\int_{D_0} |y - \bar{y}_t|^2 e^{\frac{-|x-y|^2}{2t}} d\pi_0(y)}{\hat{p}} = \frac{|y_1 - y_2|^2}{4},$$

leading to

$$-\Delta \bar{q}(t, x) \geq \frac{C_x}{t^2} \quad \text{for } t \in (0, 1]. \qquad \square$$

## C.7 Proof of well-poseness results

**Lemma C.4.** *Given $T > 0$, suppose that $F = F(t, x) \in C([0, T] \times \mathbb{R}^n, \mathbb{R}^n)$ satisfies that $F$ is locally Lipschitz continuous in $x$ variable, i.e., for any $M > 0$, there exists a constant $L_M$ such that*

$$|F(t, x) - F(t, y)| \leq L_M |x - y| \quad \text{for } x, y \in B_M(0) \text{ and } t \in [0, T]$$

*and*

$$|F(t, x)| \leq C(|x| + 1). \quad \text{for } (t, x) \in [0, T] \times \mathbb{R}^n.$$

*for a positive constant C. Then for any $x_0 \in \mathbb{R}^n$, the following equation has a unique solution*

$$\begin{cases} \dot{X}(t) = F(t, X(t)) & t \in [0, T] \\ X(0) = x_0. \end{cases}$$

Proof: The uniqueness follows from standard ODE theory. We just need to establish the global existence. Let $w(t) = |X(t)|^2$. Then

$$\dot{w}(t) \leq C_1 w(t) + C_2$$

for two positive constants $C_1$ and $C_2$ depending only on $C$. Hence for all $t \geq 0$,

$$e^{-C_1 t} w(t) \leq |x_0|^2 + \frac{C_2}{C_1} \left(1 - e^{-C_1 t}\right).$$

Hence the solution can be extended to $T$. $\qquad \square$

**Proof of Proposition 4.1** It suffices to notice that for each fixed sample $\omega$, $Y(t) = Y(t, \omega) = X_t(\omega) - W_t(\omega)$ just satisfies the regular ODE for any fixed sample

$$\begin{cases} dY(t) = F(t, Y + W(t))dt & t \in [0, \infty) \\ Y(0) = x_0. \end{cases}$$

Hence the Corollary follows from Theorem C.4 and the well known fact that $W_t(\omega) \in C([0, T], \mathbb{R}^n)$ for a.e. $\omega$. □

**Proof of Theorem 4.3** Note that $\log p(t, x)$ is a smooth function for $t > 0$, hence $\nabla \log p$ is locally Lipschitz continuous in $x$. Owing to Proposition 4.1, it suffices to show that for $q = -\log p(t, x)$,

$$|\nabla q(t, x)| \leq C_T(|x| + 1) \quad \text{for all } (t, x) \in [0, T] \times \mathbb{R}^n$$

for a constant $C_T$ depending on $C$ and $T$. By (7), it is equivalent to showing that

$$|\nabla \bar{q}(t, x)| \leq C_T(|x| + 1) \quad \text{for all } (t, x) \in [0, 1 - e^{-T}] \times \mathbb{R}^n.$$

for a constant $C_T$ depending on $C$ and $T$, which follows from Theorem 3.5. □

## C.8 Proof of Theorem 4.6

The key ingredient is the following estimates on truncation error.

**Theorem C.5.** *Assume $||D^2 g(x)||_2 \leq L$. Suppose that Assumption 2.4 holds and there exists $C_0 > 0$ such that, $\alpha_1 \leq C_0 n$ and $|\nabla g(x_0)| \leq C\sqrt{n}$, we have that for fixed $T \leq 1$ and $t_k = \frac{kT}{N}$,*

$$\sum_{i=1}^{N} \int_{t_{k-1}}^{t_k} \mathbb{E}||\nabla \bar{q}(t_k, x(t_k)) - \nabla \bar{q}(t, x(t))||^2 \, dt \leq \frac{CL^6 Tn(n \log n)^2}{N}. \tag{43}$$

*Here $C$ is a constant independent of $n$ and $L$.*

**Proof:** It suffices to show that for $s > t \in [0, T]$,

$$\mathbb{E}||\nabla \bar{q}(s, x(s)) - \nabla \bar{q}(t, x(t))||^2 \leq CL^6 (s - t)n^3 \log n$$

According to Lemma C.6 in Chen et al. (2023), it suffices to show that

$$\mathbb{E}||\nabla \bar{q}(t, x(t) + z) - \nabla \bar{q}(t, x(t))||^2 \leq Cn^2(s - t). \tag{44}$$

Here $z \sim \mathcal{N}(0, C(s - t))$.

Owing to (14) of Theorem 3.5 in our paper and $\max\{a, b\} \leq a + b$,

$$||D^2 \bar{q}(t, x)|| \leq C(|x|^2 + n \log n),$$

where $C$ depends on $(L, C_0, \alpha_2)$. See (11) for the definition of the spectral norm $|| \cdot ||_2$ of $n \times n$ matrix. Then

$$||\nabla \bar{q}(t, x(t) + z) - \nabla \bar{q}(t, x(t))||^2 \leq C(1 + |x(t)|^4 + |z|^4 + (n \log n)^2)|z|^2.$$

Note that $\mathbb{E}(z^2) \leq Cn(s - t)$ and $\mathbb{E}(z^4) \leq Cn^2(s - t)^2$. Moreover, by Cauchy inequality

$$\mathbb{E}(|x(t)|^4 z^2) \leq \sqrt{(\mathbb{E}(x^8(t))\mathbb{E}(z^4)} \leq Cn(n \log n)^2(s - t).$$

The last inequality is due to $\mathbb{E}_{p(t,x)}(x(t)^8) \leq C(n \log n)^4$ from Remark. Hence (44)holds. □

**Remark C.6.** *In the proof of Theorem C.5, instead of using Lemma C.6 in Chen et al. (2023), we may also use (15) from Theorem 3.5 to bound the difference between time,*

$$\mathbb{E}||\nabla \bar{q}(s, x(t)) - \nabla \bar{q}(t, x(t))||^2 \leq C(s - t)^2 n^4 (\log n)^3.$$

*This will lead to an extra term $\frac{Cn^4(\log n)^3}{N^2}$ on the right hand side of (43). The proof is similar. Note that when $N = \mathcal{O}(n^2)$, $\frac{n^4(\log n)^3}{N^2} \preceq \frac{n^3(\log n)^2}{N}$.*

**Remark C.7.** *We are aware of the difference of $\nabla \log p$ and $\nabla \bar{q}$ due to the translation (7), while our Lipschitz estimate is uniform in time, hence similar results of Theorem C.5 holds for $\nabla \log p$.*

## C.9 Convergence bounds under compact support manifold assumption

**Theorem C.8.** *We assume* $\text{supp}(p_0) = D_0 \subset \overline{B_M(0)}$ *and the density is smoothly defined on* $D_0$. *With early stopping* $\delta > 0$, *Let* $\hat{Q}_{T-\delta}$ *be distribution generated by uniform discretization of the exponential integrator scheme* (4), *with an approximated score satisfying Assumption 2.1.*

*If* $L_1 > 0$

$$\text{KL}(P_\delta \| \hat{Q}_{T-\delta}) \lesssim (M_2 + d)e^{-T} + T\epsilon_0^2 + \frac{dT^2 L_\delta^2}{N},$$

*where* $L_\delta = 1 + \frac{1}{\delta} + \frac{M^2}{\delta^2}$.

**Proof:** $L_\delta$ is computed from (16). Then Proposition A.3 is applied. □

## C.10 Sketch of proof of Wasserstein distance bound

Here we provide a sketch proof to a Wasserstein distance bound with full details left in a future publication. A key ingredient is to estimate the backward process $\tilde{X}_t$ in (2) and its discretized approximation $\hat{x}_t$ in (4). The two processes are coupled by the same Brownian path and initial value, hence,

$$\frac{d\|\tilde{X}_t - \hat{x}_t\|}{dt} = \frac{1}{2}\|\tilde{X}_t - \hat{x}_t\| + \frac{1}{\|\tilde{X}_t - \hat{x}_t\|}\langle \tilde{X}_t - \hat{x}_t, \nabla \log p(T-t, \tilde{X}_t) - s_\theta(T - t'_k, \hat{x}_{t'_k})\rangle$$

$$\leqslant \frac{1}{2}\|\tilde{X}_t - \hat{x}_t\| + \|\nabla \log p(T-t, \tilde{X}_t) - s_\theta(T - t'_k, \hat{x}_{t'_k})\|, \tag{45}$$

where $t \in [t'_k, t'_{k+1}]$. Then we turn to the inequality,

$$\|\nabla \log p(T-t, \tilde{X}_t) - s_\theta(T - t'_k, \hat{x}_{t'_k})\|$$

$$\leqslant \|\nabla \log p(T-t, \tilde{X}_t) - \nabla \log p(T - t'_k, \tilde{X}_{t'_k})\| + \|\nabla \log p(T - t'_k, \tilde{X}_{t'_k}) - \nabla \log p(T - t'_k, \hat{x}_{t'_k})\|$$

$$+ \|\nabla \log p(T - t'_k, \hat{x}_{t'_k}) - s_\theta(T - t'_k, \hat{x}_{t'_k})\|, \tag{46}$$

where the last term on the right hand side of (46) relates to the approximation error of the score. With the Lipschitz bound (Theorem 3.5) in hand, we estimate the first two terms, while noticing that the Lipschitz constant grows linearly while the diffusion process $\tilde{X}_t$ has exponential tail.

The bound for $E\|\tilde{X}_T - \hat{x}_T\|$ follows by taking expectation of (45) and using a Gronwall type inequality, thus implying a bound on the Wasserstein distance $W^2(\text{Law}(x_T), \text{Law}(\hat{x}_T))$. The bound on $W^2(\text{Law}(\tilde{X}_T), P_0)$ then follows from a stability analysis of $\tilde{X}$ with respect to the initial distribution, per standard arguments as in Bortoli (2022); Chen et al. (2023).

# D Broader Impact

Diffusion model is one of the most influential generative models in the AI era. Our theory gives theoretical guarantee of the lifespan of diffusion model with minimal assumption of data distribution. We discovered a theoretical characterization in the point-wise sense (stronger than prior works) on the singular behavior near generation time related to the manifold hypothesis. This provides insight for model parameterization and convergence rate improvement in practical implementations.

