# OpenReview forum: "Global Well-posedness and Convergence Analysis of Score-based Generative Models via Sharp Lipschitz Estimates"
_ICLR.cc/2025/Conference — ICLR 2025 Poster_

### Official Review · Reviewer_5LpM · 2024-11-01

**Soundness:** 3
**Presentation:** 3
**Contribution:** 3
**Rating:** 6
**Confidence:** 3

**Summary:**

The paper focuses on providing the Lipschitz estimates for the score-function in the score-based generative models. It also provides a sharp example showing that the loss of Lipschitz bound of the score function as time gets large even with nice initial data distribution.  As a by-product, well-posedness and convergence are obtained by following existing literature.

**Strengths:**

(1) Estimating the Lipschitz constant for the score-function using the idea of the PDE approach is novel and makes a nice contribution to the theory of score-based generative models.

(2) An example is provided to show the Lipschitz estimate is sharp in some sense.

**Weaknesses:**

(1) I think in the abstract, the paper overclaims that the results are obtained under minimal general assumptions of initial data because when you look at Theorem 3.1., Corollary 3.2. etc., some relatively strong assumptions are used.

(2) Well-posedness follows immediately from the estimate on the score function and Theorem 4.1. can be found in any textbook on stochastic calculus. Moreover, the convergence analysis follows immediately from existing literature (Chen et al. 2023). In that regard,
I think the main contributions of the paper are simply the Lipschitz estimates for the score function, and well-posedness and convergence analysis are not the main contributions of the paper.

**Questions:**

(1) In the last paragraph of page 1, well-poseness should be well-posedness.

(2) In the last paragraph of Section 1, Planc should be Planck.

(3) On page 3, "In the analysis, we assume an $\epsilon^{2}$ bounds for this estimation'' I think you may replace $\epsilon$ by $\epsilon_{0}$ because this is what you used in Assumption 2.1.

(4) It would be nice to add some discussions about Assumption 2.3., how it is related to the assumptions in the previous theoretical work on score-based generative models.

(5) In the statement of Corollary 3.2, $p_{0}\in C^{2}(\mathbb{R}^{d})$ but you also use $\sup_{x\in\mathbb{R}^{n}}D^{2}\log p_{0}(x)$. I think in your case $d=n$? Similar issue appears in the statement of Theorem 4.3. and Theorem 4.4., where you have used both $d$ and $n$.

(6) In Theorem 4.3., in the upper bound for $\mathrm{KL}(p_{0}\Vert\hat{q}_{T})$, the first term is $(M_{2}+d)e^{-T}$. Can you remind the readers what is $M_{2}$ here (or where it is defined)? Also, for $H_{T}$, does it have an explicit formula (depending on $L_{0},L_{1},T$) and if so you should mention it. Also, I think it will be nice to spell out the dependence on $T$ because your upper bound is written in a way with an emphasis on the dependence on $T$, and if $H_{T}$ is implicit in $T$ it will make your bound less
valuable.

(7) In order to have $\mathrm{KL}(p_{0}\Vert\hat{q}_{T})$ to be small in  the upper bound for $\mathrm{KL}(p_{0}\Vert\hat{q}_{T})$ in Theorem 4.3., it seems you have to choose $T$ to be large, but on the other hand $T<-\log(1-\frac{1}{L_{0}+1})$, which seems
problematic to me.

(8) If my understanding is correct, the main contribution of the paper is the Lipschitz estimate on the score function, rather than the convergence analysis, which follows from Chen et al. (2023) etc. from the literature. In that spirit, does that mean your Lipschitz estimate on the score function can also be used to obtain convergence analysis for probability flow ODEs by leveraging existing works on the convergence analysis of the probability flow ODEs in the literature?

---

> ### Author Response · Authors · 2024-11-21
>
> **In response to weakness part:**
>
> 1. Our main Hessian bound result for smooth initial data, Theorem 3.5, is based on Assumption 2.3, which is a typical (though not strictly necessary) technical condition in the case of smooth initial data to ensure the existence of solutions up to time $t = 1$. Theorem 3.1 and the subsequent Corollary 3.2, similar to results in previous works as noted in Remark 3.3, are included for  reader's convenience and serve as preparation for another key result in this paper, Theorem 3.4, which demonstrates the sharpness of the threshold time given in Theorem 3.1 and Corollary 3.2.
> That said, we agree with the referee that the phrase 'minimal general assumption' could be controversial. Therefore, we have removed the term 'minimal'.
>
> 2. Indeed well-posedness and convergence of numerical scheme would normally follow from the Lipschitz bound. However, there is a rewarding bonus from our work here.
> The convergence estimates for smooth initial data in Chen et al. (2023) rely on two key elements: (1) a short-time pointwise uniform Lipschitz bound, and (2) an average bound on the Hessian for longer times.
>  In contrast, our approach  leverages a pointwise Lipschitz bound that is local in space but global in time, and so is applicable to
> the more general case, including potential Wasserstein bound which is sketched in Section C.10.
>
> **In response to questions:**
>
> 1.-3. Thanks for pointing out the typo. We have corrected it and conduct a deeper proofreading to fix the notations.
>
> 4. First we would point out that our Assumption 2.3 is just to assume the tail of the distribution is upper bounded by some Gaussian. For smooth initial data, Assumption 2.3 represents a typical (though not strictly necessary) condition to ensure the existence of solutions of Fokker-Planck equation of the forward process. We added a line before Assumption 2.3 to explain this.
>
> 5. Thanks for pointing out the notation problem. In the updated submission, we use $n$ to denote the total number of dimension.
>
> 6. We have rewritten Theorem 4.3 and provided definition (or related equation number) of the relating coefficients.
>
> 7.  We agree that under the triangle inequality (Proposition A.3, or the one in Chen et al. (2023)) approach for KL bound, all complexity bounds with the same error level, will require $T=\mathcal{O}(\log n)$. So we have  included a new Remark 4.4 after  Theorem 4.3 to state the assumed regime of the near linear $\mathcal{O}(n\log n^2)$ complexity bound.
>
> 8.  Indeed, we are working on the Lipschitz bound of the score function based on probability flow ODEs.  New analytical tools, similar to Proposition A.3 or Theorem 2.1 of Chen er al. 2023, need to be developed for convergence and complexity bound. Should the reviewer point out other tools to this end, we are happy to cite them in our manuscript. We will complete this study in a future research. In the updated manuscript,  we give a sketch of proof on Wasserstein distance bound in Section C.10 of the Appendix section based on the point-wise estimates obtained here.

---

### Official Review · Reviewer_5jve · 2024-11-02

**Soundness:** 3
**Presentation:** 3
**Contribution:** 3
**Rating:** 6
**Confidence:** 3

**Summary:**

This paper addresses the global well-posedness and convergence of score-based generative models (SGMs) by providing sharp Lipschitz estimates for the score function under both smooth and non-smooth initial conditions. By developing optimal point-wise gradient and Hessian estimates, the authors establish conditions under which the backward diffusion process is well-posed up to time zero, effectively covering cases where previous approaches failed due to exploding scores or separated time regimes.

**Strengths:**

The paper introduces new sharp Lipschitz bounds for the score function, which enable global well-posedness and convergence without requiring separated time scales. This is particularly valuable for SGMs where score explosion has previously posed challenges.

**Weaknesses:**

I did not identify any major weaknesses in this paper. I listed some typos below.

Typos:
1. On the LHS of equation (3), I believe $p_t$ means $\partial_t p$. Using the notation $p_t$ will probably make the reader confused as $p_0$ denotes the initial data;
2. In line 108, it should be $Q_{t'}$ instead of $Q_t'$;
3. In line 127, the LHS should be $\hat{x}$ instead of $d\hat{x}$;
4. In line 218, it should be $\mathbb{R}^n$ instead of $\mathbb{R}^d$.

**Questions:**

1. It appears that the main assumption in this work is Assumption 2.3, which requires that the tail distribution is bounded by a Gaussian distribution. Could you provide some intuition behind this assumption? It seems potentially quite strong, as it may imply that the data is approximately Gaussian—perhaps even stronger than the strong dissipativity condition. Could you clarify why this is a reasonable assumption for real-world data distributions?

2. The statements in Section 3.3 appear somewhat inconsistent. In Theorem 3.9, it is assumed that the density is smoothly defined on $D_0$, which may be a low-dimensional manifold, while in the proof, the integral over $\mathbb{R}^n$ is equal to the integral over $D_0$. How is $d\pi_0(y)$ defined to ensure this equality holds? Is $p_0$ a density on $D_0$ or $\mathbb{R}^n$? I may have misunderstood, so please correct me if I'm mistaken.

---

> ### Author Response · Authors · 2024-11-21
>
> Thanks for the careful reading, we have fixed the typo mentioned in the review and finished rounds of proofreading to fix the notations.  Below we list our response to the questions.
>
> 1. This is a very reasonable question. First we would point out that our Assumption 2.3 is just to assume the tail of the distribution is upper bounded by some Gaussian, in particular, a distribution with bounded support follows such a assumption. For smooth initial data, Assumption 2.3 represents a typical (though not strictly necessary) condition to ensure the existence of solutions. We added a line before Assumption 2.3 to explain this.
>
> In addition, non-smooth initial data are more common in real-world applications. However, the $O({1/t})$ blow up in  (1) of inequalities (13) (Theorem 3.9 in the previous version) indicates that the denoising process can only be traced back to a certain $t > 0$, which essentially corresponds to a Gaussian-like initial data.
>
>
> 2. Thank you for pointing this out. Indeed, the assumption that "the density is smoothly defined on $D_0$" is not necessary. We only require that the initial distribution $p_0$ is a probability density supported on $D_0$, as can be seen from the proof. This assumption has been removed. Additionally, following the suggestion of another referee, we have chosen to state the estimates in (13) directly in the revised version rather than formulating them as a theorem.

---

> > ### Comment · Reviewer_5jve · 2024-11-25
> >
> > Thank you for the explanation. I will keep my score.

---

> > > ### Author Response · Authors · 2024-11-26
> > >
> > > Thanks very much for reviewing our submission and responses. Please be free to let us know if any further clarification or improvement is needed during the discussion period.

---

### Official Review · Reviewer_QYfP · 2024-11-02

**Soundness:** 2
**Presentation:** 2
**Contribution:** 2
**Rating:** 6
**Confidence:** 2

**Summary:**

Recent theoretical works on analyzing diffusion models require bounds on the Lipschitzness of the score function of the noised distribution at all time-steps. This work gives bounds on Lipschitzness under various conditions on the initial data distribution, and uses some of these bounds to instantiate prior guarantees for diffusion models.
- Theorems 3.1 and 3.4 show that if the initial data distribution is smooth, then the score of the noised distribution is Lipschitz up to some finite time T, and that this cannot be extended to all times
- Theorem 3.5 gives local Lipschitzness estimates of the score function under local assumptions
- Theorems 3.9/3.10 and Example 3.11 show that if the data distribution is supported on a manifold, then one gets Lipschitzness bounds of 1/t^2 at time t, and in fact almost everywhere one gets 1/t, and there is an example where the "almost everywhere" is necessary
- Theorem 4.3/4.4 instantiate diffusion model guarantees using Theorem 3.1 and Theorem 3.5 respectively

**Strengths:**

The paper addresses an interesting question of trying to substantiate the Lipschitzness assumptions on score functions. The result on manifolds showing that 1/t^2 can be almost-everywhere improved to 1/t seems qualitatively new and interesting, and the example showing that this is tight provides a fairly complete story.

**Weaknesses:**

The paper could be written better, as far as comparison to prior work, clarity about which of the results are most "important", readable theorem statements, and discussion about the assumptions.
- Theorem 3.1, 3.9, and 4.1 seem as though they may be fairly standard / similar to prior results, but there is very little discussion about what is new and what is included merely for completeness.
- Theorem 4.4 says "with same assumption as Corollary 3.6", which in turn says "under the same assumption as in Theorem 3.5". The stated assumption is uninterpretable: is it reasonable? I can't tell.
- The invocation of Theorem 3.5 in the proof (line 1320) provides no explanation as to how the complicated bound in Theorem 3.5 becomes this simple bound.

The most interesting result would be if the 1/t^2 -> 1/t improvement could be used to instantiate a better bound for diffusion models. However it appears that this is not done. The rest of the results in Section 4 just have lots of assumptions (e.g. the bound on T in Theorem 4.3) so it's not clear how applicable they are.

**Questions:**

N/A

---

> ### Author Response · Authors · 2024-11-21
>
> 1. For Theorem 3.1, we further clarify its connection to prior works (in Remark 3.3). Based on the referee’s suggestion, we add a sentence immediately before the statement of Theorem 3.1 to demonstrate that the associated time threshold is sharp (Theorem 3.4), which is new.
>
> Regarding Theorem 3.9 in the previous version, we have followed the referee's advice by presenting it, in the revised version,  as a standard result (16)  rather than formulating it as a theorem. As discussed in our paper, the key focus is on whether these standard estimates can be further improved. We also refined the statements following (16) to emphasize two key points: (1) the $1/t$ gradient bound is generally sharp and cannot be improved, and (2) the Hessian bound can be improved from $1/t^2$ to $1/t$ for ``most"  $x$. For the reader's convenience, we kept the proof of (16), as it is straightforward and some steps, such as (32) in the previous version, are used for the proof of Theorem 3.10 (Theorem 3.9 in the updated version).
>
> Regarding Theorem 4.1, we noted in Remark 4.2 in the previous version that it is a special case of more general results (e.g.,  Chapter IV of Ikeda $\&$ Watanabe (2014)). Since uniform Lipschitz continuity is often assumed as a standard condition for long time existence of solutions for convenience, we believe it would be beneficial for readers, especially non-experts, to explicitly state this special case as a theorem and provide its proof,  where uniform Lipschitz continuity may not hold. Thanks to the referee's suggestion, we have added several sentences just before the statement of Theorem 4.1 and in Remark 4.2 to emphasize that this is a special case of known results.
>
>
> 2. We apologize for the inconvenience during your reading, and have revised to list the assumptions in Section 2.2.  Regarding whether those technical assumptions are reasonable, for smooth initial data, Assumption 2.3 represents a typical (though not strictly necessary) condition to ensure the existence of solutions. Also, if we assume $D^2g$ is bounded, $g$ grows at most quadratically and  $Dg$ grows at most linearly. Hence it is reasonable to expect that $\alpha_1$ behaves similar to $|x_0|^2\sim O(n)$ and $\nabla g(x_0)$ behaves similar to $|x_0|\sim O(\sqrt{n})$.  From the computational point of new, the main issue is to track dependence on the dimension $n$ that could be potentially very large.
>
> 3. The bound on line 1320 of the previous version is due to  (14) of Theorem 3.5 in the revised version ( the same as (11) of Theorem 3.5 in the previous version) and the fact that $\max(a,b)\leq a+b$ after absorbing all other constants depending on $C_0$, $\alpha_2$ and $L$ but independent of $n$.   We added this explanation in the revised version.
>
>
> 4. As mentioned in the paper, if the Hessian bound improvement ${1/t^2}\to {1/t}$ can be established for every $x$, then an improved convergence rate of the Wasserstein bound follows immediately from Theorem 3 of Bortoli (2022).  Our result shows that this Hessian bound improvement typically holds except on a set of measure zero of $x$. Accordingly,  the theoretical assumption of $O(1/t)$ Hessian bound is meaningful in most situations when addressing the convergence rate issue. We share the referee's view  that it would be ideal  if we could   quantify rigorously  how  this small exceptional set (usually inevitable for non-convex data set) could impact the convergence rate. However, this is mathematically very challenging due to the complex topological structure of the exceptional set, which we plan to investigate in the future. A sentence is added in the revised version to clarify this point.

---

> > ### Comment · Reviewer_QYfP · 2024-11-23
> >
> > Thanks for the response and revision. One follow-up question: in Remark 4.6, it's stated that "An important
> > feature of these new bounds is their uniformity in time". What does "uniformity in time" mean here?

---

> > > ### Author Response · Authors · 2024-11-24
> > >
> > > Thanks for following up.
> > >
> > > Uniform in time means that, the local Lipschitz bound we got in Theorem 3.5 and global Lipschitz bound for 'log-concave' part of Corollary 3.2, do not depend on time $t$.   Note that the bound on the right hand side of equality (14) (revised versoin) does not have t dependence for all $t\in [0,1]$. (Note here $t$ is a transformed variable, in the original equation (1), it is  $t\in [0,\infty)$)
> > >
> > > Without the Assumption 2.3 (or similary Assumption 2.4), the local bound might blow up as $t\to 1$, i.e., the bound on equality (14)  could contain terms like $1/(1-t)^r$ for some $r\geq 1$. Without assumption of Lipschitz for initial value, the Lipchitz may have singularity as $1/t$ as stated in Theorem 3.9.
> > >
> > > A direct example is by setting the initial condition to be some Gaussian whose covariance matrix is $C$ and mean is $0$. Then one can directly compute the Hessian of $\log p(t,\cdot)$ as $e^{-t}C+(1-e^{-t})I$, which is bounded (uniformly) for any $t\in [0,\infty)$. Our analysis, especially  Theorem 3.5, showed such a uniform in time bound under general assumptions, which is not limited to Gaussian or log concave distributions.

---

> > > > ### Comment · Reviewer_QYfP · 2024-11-24
> > > >
> > > > I see, thanks. I've increased my score due to the authors' clarifications and revisions. That said, I do think the paper is difficult to digest, and would benefit from some more clear identification of what are the main "take-home" results of standalone interest (e.g. perhaps Theorem 4.5?) and which results are intermediate technical results. The "main contributions" section could use hyperlink references to the relevant theorems, and Section 4 could use a brief outline of its contents at the top (similar to what Section 3 has).

---

> > > > > ### Author Response · Authors · 2024-11-25
> > > > >
> > > > > Thanks a lot for the input. We will revise soon according to your suggestion.

---

> > > > > ### Author Response · Authors · 2024-11-26
> > > > >
> > > > > We appreciate the further comments regarding the readability of the submission. We have provided an updated manuscript, including the following updates:
> > > > > 1. An updated main contribution in page 2 with links to theorems.
> > > > > 2. Bond font in the paragraph before section 2, indicating the main theoretical results are the estimates of Hessian of score potential function, namely Lipschitz estimates of the score.
> > > > > 3. A paragraph at the start of Section 4 outlining the contents.
> > > > > 4. An updated conclusion section, including short descriptions of theorems (with hyperlinks) and their implications.
> > > > >
> > > > > Thanks again for the comments, please be free to let us know if any further clarification or improvement is needed during the discussion period.

---

### Official Review · Reviewer_vaeP · 2024-11-02

**Soundness:** 4
**Presentation:** 3
**Contribution:** 3
**Rating:** 6
**Confidence:** 3

**Summary:**

This paper investigates the mathematical framework behind score-based generative models (SGMs). The authors' primary focus is to establish conditions under which these models are well-posed and convergent.

Key points from the paper include:
1. **Well-Posedness of SGMs:** The authors prove that SGMs are well-posed under global conditions. They introduce Lipschitz estimates that maintain stability and prevent the “blow-up” of the score function, which has been an issue in practical applications where score gradients become unstable.
2. **Convergence and Sharp Lipschitz Bounds:** They provide a Lipschitz bound for the score function, addressing both smooth and non-smooth cases. This bound helps establish global convergence of the backward process (used in sample generation), even for non-log-concave distributions. For non-smooth data distributions, the bound asymptotically scales as $1/t$, ensuring convergence on low-dimensional manifolds with compact support.
3. **Theoretical Contributions:** The authors introduce rigorous estimates for the Hessian and gradient of the score function, reinforcing the stability and convergence properties. This allows the generative models to better approximate complex data distributions.

The paper concludes with insights into how these findings could guide model implementation, such as early stopping strategies and parameter adjustments, to ensure stable and efficient generation in real-world applications.

**Strengths:**

The major strengths of this paper are:
- Establishing crucial bounds on the Hessian of the score function, which significantly enhance existing bounds on the KL divergence from *Chen et al. (2023)*. These bounds are vital for assessing the fidelity of generated distributions.
- Carefully tracking the dependence on dimensionality throughout the analysis, a critical aspect for score-based generative models (SGMs) given their frequent application to high-dimensional data distributions.
- Characterizing the explosion of the score function as $t$ approaches $0$. Although this instability near zero has been commonly observed in practice, the authors quantify the explosion’s magnitude through PDE techniques, opening the door to improve model stability in high-dimensional generative tasks.
- Describing the case of compactly supported data distributions, shedding light on the theoretical limits of score-based generative models (SGMs). This perspective allows for a deeper understanding of how SGMs behave when data is restricted within a bounded region, which has practical relevance given that real-world data often exhibit such constraints.

**Weaknesses:**

Here are some weaknesses of the paper:
1. **Limited Comparison with Recent Literature:** The paper lacks sufficient engagement with recent literature on SGMs, particularly from a PDE perspective. A broader comparison with recent works is needed, especially regarding bounds on KL divergence, as several relevant contributions exist and should be cited.
2. **Redundancy in Subsection 2.2:** The computations in Subsection 2.2 are already present in the literature, as seen in *Conforti, Durmus, Gentiloni-Silveri (2023)*. While this content is valuable for readability, it might be better suited to an appendix section to streamline the main text.
3. **Unclear Novelty in Theorems 4.3 and 4.4:** The novelty of Theorems 4.3 and 4.4 is unclear in comparison to the results in *Chen et al. (2023)*. It would be beneficial to clarify how the bounds on the Hessians improve convergence rates relative to this paper, particularly in establishing any distinct contributions.
4. **Lack of Appendix Structure:** Adding a paragraph at the beginning of the appendix outlining its structure would improve navigation and clarity for readers.

Addressing these points would strengthen the paper, and I would consider raising the evaluation if these aspects are clarified and expanded.

**Questions:**

All the questions have been addressed in the *Weaknesses* section.

---

> ### Author Response · Authors · 2024-11-21
>
> **In response to Weaknesses**
>
> 1. To the authors' best knowledge, we shall include two results in the updated Remark 4.6: 1) 'Nearly -Linear Convergence Bounds for Diffusion Models via Stochastic Localization' (2024) that provides a $d$-linear complexity bound in the early stopping setup; 2) a concurrent 'KL Convergence Guarantees for Score Diffusion Models under Minimal Data Assumptions' (2024) that provides a linear in dimension bound without early stopping.
>
> 2. We agree with the referee that the material in Subsection 2.2 is well-known to experts. Nevertheless, we believe it is more convenient for readers, particularly non-experts, to include it in the main text. A sentence will be added at the beginning of subsection 2.2 to clarify this point.
>
> 3. We shall provide a comparison to results in Chen et all. (2023) as Remark 4.6 in the revised version. We agree that we have a complexity bound slightly higher that Theorem 2.5 in Chen et all. (2023) under similar condition of initial data, although both of them are polynomial in dimension. On the other hand,  since our new Hessian bound is homogeneous in time, our theorems apply to uniform discretization in time, which does not require any prior knowledge of  the target distribution and  hence is more realistic. In particular, we provide a sketch of proof about how to use those pointwise Hessian bound to obtain convergence rate under Wasserstein metric in whole space.
>
> 4. Thank you for the suggestion. We shall insert a paragraph at the beginning of the appendix summarizing the structure of the manuscript.

---

> > ### Comment · Reviewer_vaeP · 2024-11-24
> >
> > Thank you to the authors for their work in revising the submission. The changes made address key concerns and improve the overall clarity and support for the paper’s claims. Based on these revisions, I have updated my score to 6.

---

> > > ### Author Response · Authors · 2024-11-25
> > >
> > > Thanks a lot for the review and update.

---

### Author Response · Authors · 2024-11-21

We appreciate for the valuable comments by the reviewers. We have updated our submission in response and will provide point by point answers to the comments received. We temporarily mark the modification with blue color in the current submission.

---

### Meta-Review · Area_Chair_a8AU · 2024-12-20

**Metareview:**

The manuscript studies the score-based generative models. Compared with previous studies, the manuscript establishes sharper Lipschitz estimate of the score function, which in particular covers the non-smooth case when the data distribution might lie on lower-dimensional manifold. Overall, while it feels that the theoretical contribution towards understanding score-based diffusion models is marginal compared with the literature, the manuscript is solid and the authors have addressed all reviewers' concerns through the discussion phase. The reviewers unanimously recommends marginal above acceptance, and the meta-reviewer agrees with the assessment.

**Additional Comments On Reviewer Discussion:**

The authors have addressed reviewers' concerns during the discussion phase.

---

### Decision · Program_Chairs · 2025-01-22

Accept (Poster)